# TOC1–PIF4 interaction mediates the circadian gating of thermoresponsive growth in *Arabidopsis*

Jia-Ying Zhu[1,*], Eunkyoo Oh[1,2,*], Tina Wang[1] & Zhi-Yong Wang[1]

*Arabidopsis* adapts to elevated temperature by promoting stem elongation and hyponastic growth through a temperature-responsive transcription factor PIF4. Here we show that the evening-expressed clock component TOC1 interacts with and inactivates PIF4, thereby suppressing thermoresponsive growth in the evening. We find that the expression of PIF4 target genes show circadian rhythms of thermosensitivity, with minimum responsiveness in the evening when TOC1 level is high. Loss of function of TOC1 and its close homologue PRR5 restores thermosensitivity in the evening, whereas TOC1 overexpression causes thermo insensitivity, demonstrating that TOC1 mediates the evening-specific inhibition of thermo-responses. We further show that PIF4 is required for thermoadaptation mediated by moderately elevated temperature. Our results demonstrate that the interaction between TOC1 and PIF4 mediates the circadian gating of thermoresponsive growth, which may serve to increase fitness by matching thermoresponsiveness with the day–night cycles of fluctuating temperature and light conditions.

[1] Department of Plant Biology, Carnegie Institution for Science, Stanford, California 94305, USA. [2] Department of Bioenergy Science and Technology, Chonnam National University, Gwangju 61186, Korea. * These authors contributed equally to this work. Correspondence and requests for materials should be addressed to E.O. (email: eoh@jnu.ac.kr) or to Z.-Y.W. (email: zywang24@stanford.edu).

Understanding how plants adapt to elevated temperature is becoming more and more important, as heat waves have caused severe crop losses worldwide in recent years and the situation is projected to get worse with the climate continuing to warm up[1]. For many plants, a small increase in ambient temperature profoundly affects growth and development. *Arabidopsis thaliana*, similar to many higher plants, has evolved to adapt to elevated temperature by a suite of morphological changes, including increased elongation of hypocotyls and petioles, hyponastic growth and development of thinner leaves[2]. Such developmental and morphological changes induced by high ambient temperature, below the heat-stress range, is collectively named thermomorphogenesis[3]. Thermomorphogenesis is an adaptive growth that presumably reduces the damages caused by potentially detrimental high-temperature conditions[2,4], which naturally occur around the middle of summer days.

Thermomorphogenesis is mainly mediated by PHYTO-CHROME INTERACTING FACTOR4 (PIF4), one of the bHLH transcription factors that promote stem elongation[5]. Warm temperature increases *PIF4* expression and the *pif4*-null mutant is defective in the growth response to warm temperature[6,7]. PIF4 interacts with the brassinosteroid-activated transcription factor BRASSINAZOLE-RESISTANT 1 to activate a core transcriptome that drives shoot organ elongation[8,9]. PIF4 also directly activates several auxin biosynthesis genes including *YUCCA 8* (*YUC8*), to increase the endogenous auxin level, which further promotes hypocotyl elongation and hyponastic leaf growth[10,11].

PIF4 is a central growth regulator that is controlled by a wide range of environmental signals and endogenous programmes. In addition to activation by warm temperature, PIF4 is inhibited by light through phytochrome-mediated degradation and cryptochrome-mediated inactivation[5,12,13]. Further, the *PIF4* RNA expression is controlled by the circadian clock[14,15]. Consequently, both light condition and the circadian clock influence thermomorphogenesis by modulating the level of PIF4 (refs 3,13). Recent studies have shown that the transcription factor EARLY FLOWERING3 (ELF3), a component of the evening complex (EC) of the circadian clock, represses *PIF4* RNA expression and warm temperature induces *PIF4* expression by inhibiting ELF3 binding to the *PIF4* promoter[16–18]. Furthermore, *elf3*-null mutant plants grown under short-day conditions develop long hypocotyls that are insensitive to warm temperature[17,18]. These studies demonstrate that ELF3 mediates temperature regulation of *PIF4* transcription[17,18].

In this study, we show that the evening-expressed circadian clock protein TIMING OF CAB EXPRESSION1 (TOC1) directly interacts with PIF4 and inhibits its ability to activate target gene transcription, thereby suppressing thermomorphogenesis specifically at the end of day and evening. Such circadian gating makes PIF4-mediated growth more responsive to temperature during the day, when the damage by extreme heat is enhanced by strong light, than during the evening. Furthermore, we show that the PIF4 is required for increased heat tolerance mediated by prior adaptation at moderately elevated temperature. Our results demonstrate that the interaction between TOC1 and PIF4 mediates the circadian gating of thermoresponsive growth, which enhances survival of heat stress by matching adaptive growth responses with the day–night cycles of fluctuating temperature and light conditions.

## Results

**TOC1 interacts with PIF4 and inhibits PIF4 activity**. A previous study showed that TOC1 interacts with several PIF factors

including PIF4 in a yeast two-hybrid screen[14]. However, the physiological functions of the TOC1–PIF4 interactions have not been revealed. We first verified that TOC1 interacts with PIF4 by yeast two-hybrid assays (Fig. 1a–d and Supplementary Fig. 1). Domain deletion analysis further revealed that the interaction involves the carboxy-terminal domain of TOC1 (amino acid 325–533; Fig. 1c) and the bHLH domain of PIF4 (Fig. 1d). PIF4 also interacted with the TOC1 homologue PSEUDORESPONSE REGULATOR5 (PRR5), but not with other PRR family members (PRR3, PRR7 and PRR9; Supplementary Fig. 2a). We further confirmed the TOC1–PIF4 *in vivo* interaction by co-immunoprecipitation (co-IP) assay (Fig. 1e). Consistent with the TOC1–PIF4 direct interaction, about half of the TOC1 target genes that were identified in previous chromatin immunoprecipitation sequencing (ChIP-Seq) assay were also PIF4 target genes (Fig. 1f)[9,19]. Moreover, the binding sites of PIF4 and TOC1 in the shared target genes were close to each other (Fig. 1g), suggesting that TOC1 and PIF4 tend to bind to the same genomic locations. Interestingly, the motif analysis showed that the PIF4 binding motif, the G-box motif (Fig. 1h)[9], was the most enriched motif in the TOC1-binding regions in the TOC1 and PIF4 shared target genes, but it was much less enriched in the TOC1-only target genes (Fig. 1h,i). These results suggest that PIF4 and TOC1 may bind together at G-box motifs. In addition to TOC1, PRR5 also shared a large number of target genes with PIF4 and their binding sites were located close to each other (Supplementary Fig. 2b,c)[20].

To examine how the TOC1–PIF4 interaction affects PIF4 activity, we first performed a transient gene expression assay using *Arabidopsis* mesophyll protoplasts. In line with the previous study[9], *IAA19* promoter activity was increased by PIF4 (Fig. 2a). Co-transfection with TOC1 significantly repressed the PIF4 activation of *IAA19* promoter (Fig. 2a). However, PIF4 protein level was not altered in *TOC1-OX* plants compared with wild type (Fig. 2b). Moreover, ChIP assays showed that overexpression of *TOC1* did not affect PIF4 binding to *YUC8*, *IAA19* and *IAA29* promoters (Fig. 2c), indicating that the TOC1–PIF4 interaction does not interfere with PIF4 DNA-binding ability, which is consistent with the large overlap between PIF4 and TOC1 direct target genes. Thus, TOC1 appears to directly repress the PIF4 ability to activate target gene expression, which is consistent with the previous findings that the pseudoreceiver (PR) domain of TOC1 has a transcriptional repression activity[19,21]. To determine whether TOC1 can repress PIF4 activity without altering PIF4 DNA-binding ability, we carried out transcriptional reporter gene assays using a *GAL4-UAS* promoter-GUS reporter gene and a GAL4–PIF4 fusion transcription factor, which bind to the reporter promoter through the DNA-binding domain of GAL4 (Fig. 2d). The GUS activity was increased more than 14 folds by the GAL4–PIF4 fusion protein compared with a negative control, whereas co-transfection with TOC1 reduced the fold induction of GUS activity by GAL4–PIF4 (Fig. 2e). These results demonstrate that TOC1–PIF4 interaction represses the PIF4 ability to activate target gene transcription.

**TOC1 suppresses thermomorphogenesis**. As PIF4 is required for thermomorphogenesis including hypocotyl growth responses to warm temperature, it is possible that the TOC1–PIF4 inter-action suppresses thermo-responsive growth. The hypocotyls of *toc1-2* mutants were more elongated by transient high-temperature exposure than those of wild type (Fig. 3a), indicating that TOC1 represses warm temperature-induced hypocotyl growth. The thermo-hypersensitivity of the *toc1-2* mutant was abolished by *pif4* mutation (Fig. 3b), confirming that TOC1 acts upstream

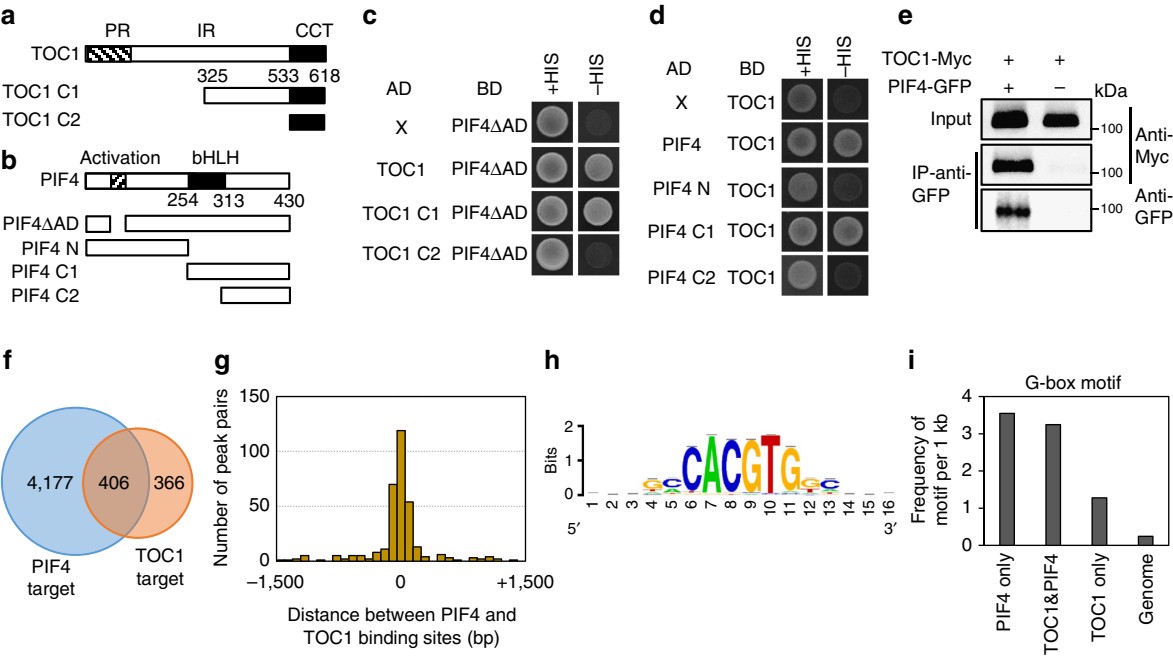

**Figure 1 | TOC1 directly interacts with PIF4. (a,b)** Box diagram of various fragments of PIF4 and TOC1 used in **c,d**. (**c,d**) Yeast two-hybrid assays. Yeast clones were grown on the synthetic dropout medium (+HIS) or synthetic dropout medium without histidine plus 1 mM 3-amino-1,2,4-Triazol (3-AT) (−HIS). The experiments were replicated with additional yeast clones (Supplementary Fig. 1). (**e**) Co-IP assays. Protein extracts from protoplasts expressing TOC1-Myc or TOC1-Myc and PIF4-GFP were immunoprecipitated with anti-GFP antibody and analysed by immunoblottings with anti-GFP or anti-Myc antibody. The molecular weight (kDa) is indicated on the right side of the gel. (**f**) Overlap between target genes of PIF4 and TOC1 identified in the previous ChIP-Seq studies[9,19] is statistically significant (Fisher's exact test $P < 2 \times 10^{-16}$). (**g**) Distribution of distances between the binding sites of PIF4 and TOC1 in their common target genes identified in **f**. (**h**) The most enriched motif in the TOC1-binding sites of the PIF4 and TOC1 common target genes. (**i**) Frequency of G-box motif per 1 kb in the PIF4-specific binding sites (PIF4 only) and TOC1-binding sites of the PIF4 and TOC1 common target genes (TOC1 & PIF4) or TOC1-specific target genes (TOC1 only).

of PIF4 in the thermo-responsive growth regulation. Consistent with the *toc1-2* phenotype, *TOC1-OX* plants were completely insensitive to a 3-day high-temperature treatment (Fig. 3c). Plants overexpressing another PIF4-interacting PRR member, PRR5 (*PRR5-OX*) were also insensitive to warm temperature (Fig. 3d). Furthermore, the *zeitlupe* (*ztl-105*) mutant, which accumulates both TOC1 and PRR5 (refs 22,23), showed a reduced sensitivity to warm temperature (Fig. 3e). These results provide genetic evidence that TOC1 and PRR5 suppress thermomorphogenesis.

To understand the molecular mechanisms of how TOC1 suppresses thermomorphogenesis, we examined the responses of PIF4 target genes to 4 h-warm-temperature treatment in the *TOC1-OX* plants. Consistent with the thermo-insensitive hypocotyl growth of *TOC1-OX*, the warm-temperature activation of PIF4 target genes, *YUC8*, *IAA19* and *IAA29*, was abolished in *TOC1-OX* (Fig. 3f and Supplementary Fig. 3). However, the expression of the early high-temperature-induced gene *HEAT SHOCK PROTEIN 70* (HSP70) was normally induced by warm temperature in *TOC1-OX* (Supplementary Fig. 4), indicating that *TOC1-OX* specifically abolished the PIF4-mediated temperature-responsive gene expression. In contrast to the PIF4 target genes, *PIF4* expression was similarly induced by warm temperature in the wild type and *TOC1-OX* (Fig. 3g). Similarly, thermo-insensitive expression of *YUC8* and *IAA29* accompanied with normal warmth activation of *PIF4* expression was observed also in the *PRR5-OX* plants (Fig. 3h,i), suggesting that TOC1 and PRR5 inhibit the warm-temperature activation of the PIF4 target genes through direct repression of the PIF4 protein activity. Furthermore, PIF4 protein stability and DNA-binding ability were not significantly affected in *TOC1-OX* (Fig. 2b,c) and ChIP assays showed TOC1 association with *IAA19*, *IAA29* and *YUC8*

promoters *in vivo* (Fig. 3j). Taken together, these results illustrate that TOC1 suppresses thermomorphogenesis by directly repressing the PIF4 activity.

As ELF3 has also been reported to directly interact with PIF4 and repress PIF4 activity by preventing PIF4 from binding to DNA[24], it is possible that TOC1 acts together with ELF3 to repress the PIF4 activity. However, the hypocotyl growth and PIF4 target gene expression (*YUC8*, *IAA19* and *IAA29*) were largely insensitive to warm temperature in the *TOC1-OX;elf3* plants, similar to that of *TOC1-OX* and in contrast to the robust responses of wild-type and *elf3* plants (Fig. 3k,l and Supplementary Fig. 3). These results suggest that TOC1 is able to inhibit the PIF4 activity in the absence of ELF3.

**The circadian clock gates PIF4-mediated thermoresponses**. Given that *TOC1* and *PRR5* levels oscillate with peaks around Zeitgeber Time (ZT) 12–14 and ZT10–12, respectively[22,23] (Supplementary Fig. 5), it is highly likely to be that accumulated TOC1 and PRR5 suppress PIF4 activity and thermomorphogenesis at the evening. To test whether thermo-activation of PIF4 activity is gated by the circadian clock at a posttranscriptional level, we examined the RNA levels of *PIF4* and its target gene *YUC8* after a short-term temperature increase (20 °C–29 °C for 4 h) given at various times during the day or night of a diurnal light/dark cycle condition (12 h light and 12 h dark). Plants treated at 29 °C during ZT0–4 showed only ∼25% increase of *PIF4* RNA level but over threefold increase of *YUC8* RNA level, compared with those maintained at 20 °C (Fig. 4a). In contrast, treatments with warm temperature during ZT8–12 and ZT12–16 increased *PIF4* RNA level by over fivefold, but did not alter *YUC8* expression level significantly (Fig. 4a).

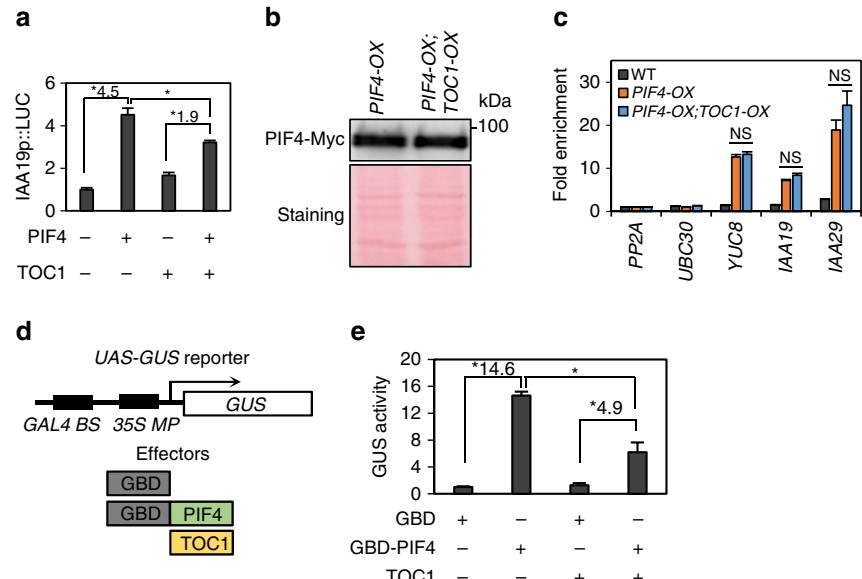

**Figure 2 | TOC1 represses PIF4 ability to activate target gene transcription.** (**a**) Transient gene expression assays. *IAA19p::Luc* was co-transfected with *PIF4-GFP, TOC1-GFP* and *35S::renilla luciferase* into *Arabidopsis* mesophyll protoplasts. Luciferase activity levels were normalized to *Renilla* luciferase activity. Error bars indicate s.d. ($n = 3$). *$P < 0.05$ (Student's *t*-test), numbers indicate fold induction by PIF4. (**b**) Western blotting with anti-Myc antibodies showing PIF4-Myc protein levels in *PIF4-OX* (PIF4-Myc) and *PIF4-OX;TOC1-OX* samples. Equal loading of samples is shown by Ponceau S staining. The molecular weight (kDa) is indicated on the right side of the gel. (**c**) ChIP-quantitative PCR assays of PIF4 binding to *YUC8, IAA19* and *IAA29* promoters. Five-day-old *35S::PIF4-MYC* or *35S::PIF4-MYC;TOC1-OX* seedlings were treated with 29 °C for 4 h and then used for ChIP assays using an anti-Myc antibody. The enrichment of DNA was normalized to that of the *PP2A* coding region. Error bars indicate s.d. ($n = 3$); NS, not significant (Student's *t*-test $P \geq 0.05$). (**d,e**) Transient gene expression assays. The *UAS-GUS* reporter construct was co-transfected with *GBD-PIF4* with or without *TOC1*, and *35S::Luc* as an internal control. The GUS reporter activities were normalized to the luciferase activity and then to the empty vector control. *GAL4 BS*, GAL4-binding site; *35S MP, 35S* minimal promoter; GBD, GAL4 DNA-binding domain. Error bars indicate s.d. ($n = 3$). *$P < 0.05$ (Student's *t*-test), numbers indicate fold induction (GBD-PIF4/GBD).

Comparable increases of *PIF4* and *YUC8* RNA levels were only observed during ZT16–20 and ZT20–24 (Fig. 4a). The patterns of *PIF4* expression suggest that an evening-specific factor that inhibits *PIF4* RNA expression is inactivated by warm temperature, consistent with the role reported for ELF3 (refs 17,18,25). The thermo-insensitivity of *YUC8* during ZT8–16, when the *PIF4* RNA level is increased dramatically by warm temperature, is consistent with an evening-specific factor such as TOC1 preventing PIF4 from activating its target genes. The decrease of TOC1 level late at night presumably allows *YUC8* activation by increased PIF4 level in the warm temperature-treated plants, consistent with previous observation of strong thermoresponsive growth at the end of night under diurnal conditions[17,18].

We next accessed the circadian clock effects on the PIF4-mediated thermo-responses without confounding effects of light/dark cycles, by analysing the responses of *PIF4* and *YUC8* to warm temperature under continuous light after entrainment with 4 days of 12 h light and 12 h dark cycles at 20 °C. In such a free running clock condition, *PIF4* expression level showed a circadian rhythm, with a high level in ZT4–8 and a low level in ZT12–20 (Fig. 4b), which is consistent with previous reports[14,26]. Similar to observations under diurnal conditions (Fig. 4a), each 4 h-warm-temperature treatment (29 °C) given during ZT8–20 increased *PIF4* RNA levels but caused no significant changes of *YUC8* RNA level (Fig. 4b). In contrast, during ZT0–8, the 4 h warm-temperature treatment had little effect on *PIF4* RNA level but increased the *YUC8* RNA level more than twofolds (Fig. 4b). The expression levels of additional PIF4 target genes, *IAA19* and *IAA29*, showed similar patterns of warm-temperature responses to *YUC8*, distinct from that of *PIF4* RNA level (supplementary Fig. 6). In contrast, *HSP 70* expression was highly

induced by warm temperature at all ZTs (Supplementary Fig. 6), suggesting that the clock does not affect the primary temperature perception. The distinct warm temperature responsiveness of the transcript levels of *PIF4* and PIF4 target genes at different circadian time under free-running conditions further support that the circadian clock specifically regulates PIF4 at not only the RNA level but also the posttranslational level. In particular, the warm-temperature induction of *PIF4* RNA and warm-temperature insensitivity of PIF4 target genes in the evening is consistent with the heat alleviation of the ELF3-mediated transcriptional repression and the TOC1-mediated posttranslational inhibition of PIF4, respectively, whereas the dramatic target gene response without *PIF4* RNA response suggests posttranslational activation of PIF4 by warm temperature.

A previous study reported that PIF4 protein levels are higher at 25 °C than at 15 °C under blue and red photoperiods[27]. We thus tested whether warm temperature increases the accumulation of PIF4 protein at the posttranslational level under our growth conditions. Immunoblotting analysis of PIF4-Myc expressed from its native promoter showed an increase of PIF4 accumulation by warm temperature at both ZT0–4 and ZT12–16 (Fig. 4c). Furthermore, PIF4-Myc expressed from the constitutive *35S* promoter was also increased by warm temperature (Fig. 4d). Consistent with the increase of PIF4 protein levels, ChIP assays showed that PIF4 binding to its target promoters was significantly increased by warm temperature at all circadian time examined, except for *YUC8* at ZT0–4 (Fig. 4e). These results confirm that warm temperature also increases PIF4 level through a posttranscriptional mechanism independent of the clock regulation of *PIF4* RNA, whereas the transcription activation activity of PIF4 is inhibited by the circadian clock in the evening when the levels of TOC1 and PRR5 are high.

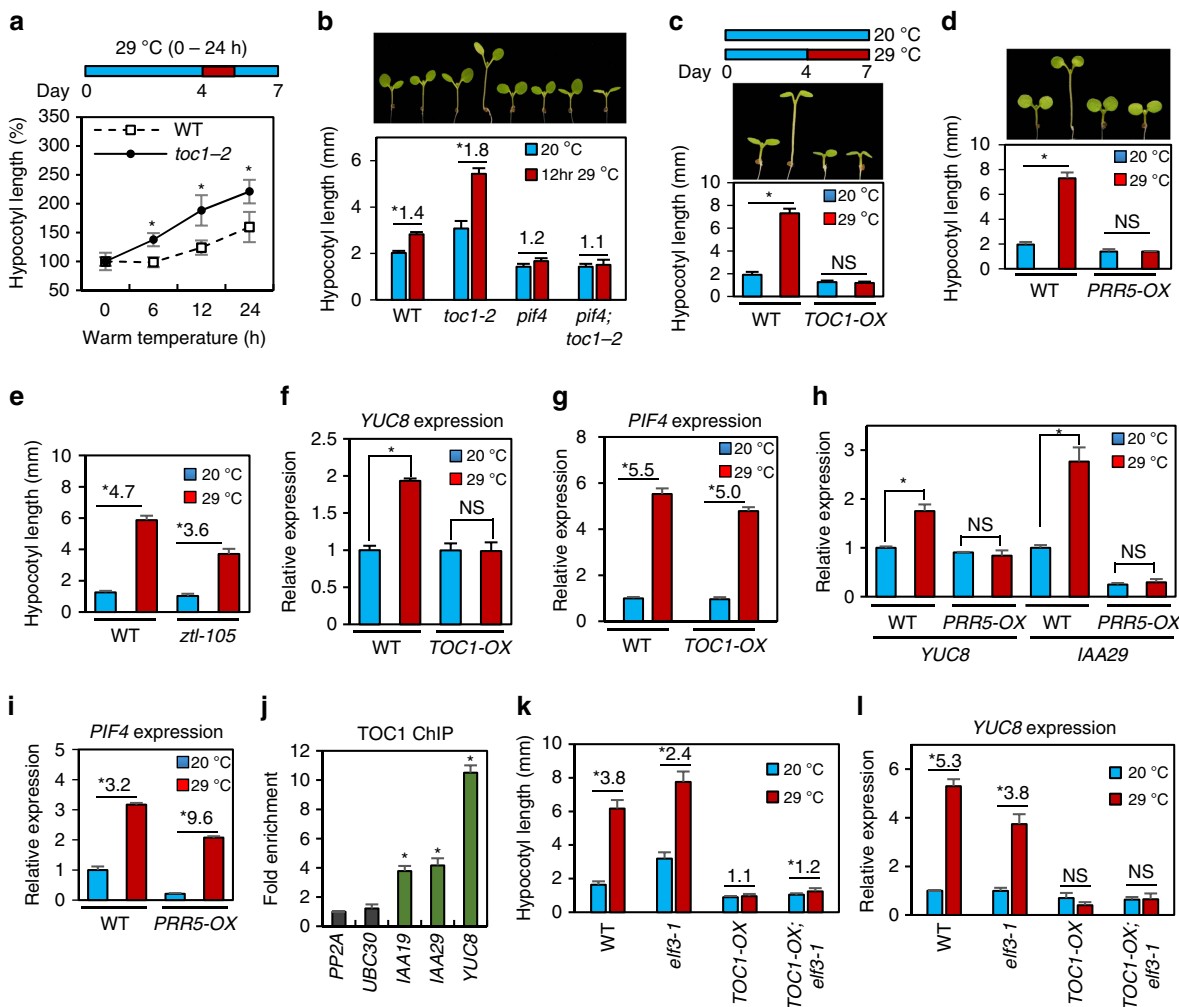

**Figure 3 | TOC1 suppresses thermomorphogenesis and warm-temperature activation of PIF4 target genes.** (**a,b**) Hypocotyl length of seedlings grown under the continuous white light at 20 °C for 4 days followed by the indicated time of warm temperature (29 °C) treatment (**a**) or 12 h-warm-temperature treatment (**b**) and then grown at 20 °C till hypocotyl measurement on the seventh day. Error bars in **a–e** indicate s.d. (n = 10 plants). *P < 0.05 (Student's t-test). Numbers in **b** indicate ratios of hypocotyl lengths (29 °C/20 °C). (**c–e**) Hypocotyl lengths of wild-type (WT), TOC1-OX, PRR5-OX and ztl-105 seedlings grown under the continuous white light at 20 °C for 7 days or 20 °C for 4 days followed by 29 °C for 3 days. *P < 0.05 (Student's t-test); NS, not significant (P ≥ 0.05); numbers in **e** indicate ratio of hypocotyl lengths (29 °C/20 °C). (**f,g**) qRT–PCR analysis of YUC8 (**f**) and PIF4 (**g**) expression in WT and TOC1-OX seedlings grown at 20 °C for 5 days then incubated at 20 °C or 29 °C for 4 h. Gene expression levels were normalized to PP2A and presented as values relative to that of wild type at 20 °C. Error bars in **f–i** indicate s.d. (n = 3). *P < 0.05 (Student's t-test); NS, not significant (P ≥ 0.05); numbers in **g** indicate ratio of PIF4 expression levels (29 °C/20 °C). (**h,i**) qRT–PCR analysis of the expression levels of PIF4 and its target genes YUC8 and IAA29 in WT and PRR5-OX seedlings grown at 20 °C for 5 days then incubated at 20 °C or 29 °C for 4 h. *P < 0.05 (Student's t-test); NS, not significant (P ≥ 0.05); numbers in **i** indicate ratio of PIF4 expression levels (29 °C/20 °C). (**j**) TOC1 ChIP assays showing TOC1 binds to the promoters of PIF4 target genes. Five-day-old TOC1p::TOC1-YFP seedlings were used for ChIP assay. Enrichment of DNA was calculated as the ratio between TOC1p::TOC1-YFP and WT control, normalized to that of the PP2A coding region as an internal reference. Error bars indicate s.d. (n = 3). *P < 0.05 (Student's t-test). (**k**) Hypocotyl lengths of TOC1-OX;elf3-1 seedlings grown under continuous white light at 20 °C for 7 days or 20 °C for 4 days followed by 29 °C for 3 days. Error bars indicate s.d. (n = 10 plants). *P < 0.05 (Student's t-test); numbers indicate ratio of hypocotyl lengths (29 °C/20 °C). (**l**) qRT–PCR analysis of YUC8 expression in seedlings grown at 20 °C for 5 days then incubated at 20 °C or 29 °C for 4 h. *P < 0.05 (Student's t-test); NS, not significant (P ≥ 0.05); numbers indicate ratio of YUC8 expression (29 °C/20 °C).

**TOC1/PRR5 mediate the circadian gating of thermoresponses.** We next investigated whether TOC1 and PRR5 are required for clock gating, and specifically the inhibition of thermosensitivity at the evening. To test this hypothesis, we analysed the warm-temperature responses of PIF4 and YUC8 expression at different circadian time in toc1 and toc1;prr5 mutants. At 20 °C, PIF4 expression was repressed at the subjective early night (ZT12–16), but it was restored by 4 h-warm-temperature treatment (Fig. 5a). However, the PIF4 expression was partly de-repressed in the toc1 and toc1;prr5 mutants at ZT12 and ZT16 (Fig. 5a), indicating that TOC1 and PRR5 redundantly repress PIF4 expression at

the early night. Previous ChIP-Seq analyses have shown that both TOC1 and PRR5 bind to the PIF4 promoter[19,20]. In addition, the PIF4 promoter contains the evening element-like-expanded (EE-like-expanded) motif (Supplementary Fig. 7a) that was previously identified as a TOC1 DNA-binding motif[19]. Our ChIP assays confirmed that TOC1 directly bound to the PIF4 promoter region containing the EE-like-expanded motif (Supplementary Fig. 7b), indicating that TOC1 directly represses PIF4 expression at the early night.

Although PIF4 expression increased dramatically after wild-type plants were treated with warm temperature during ZT8–12

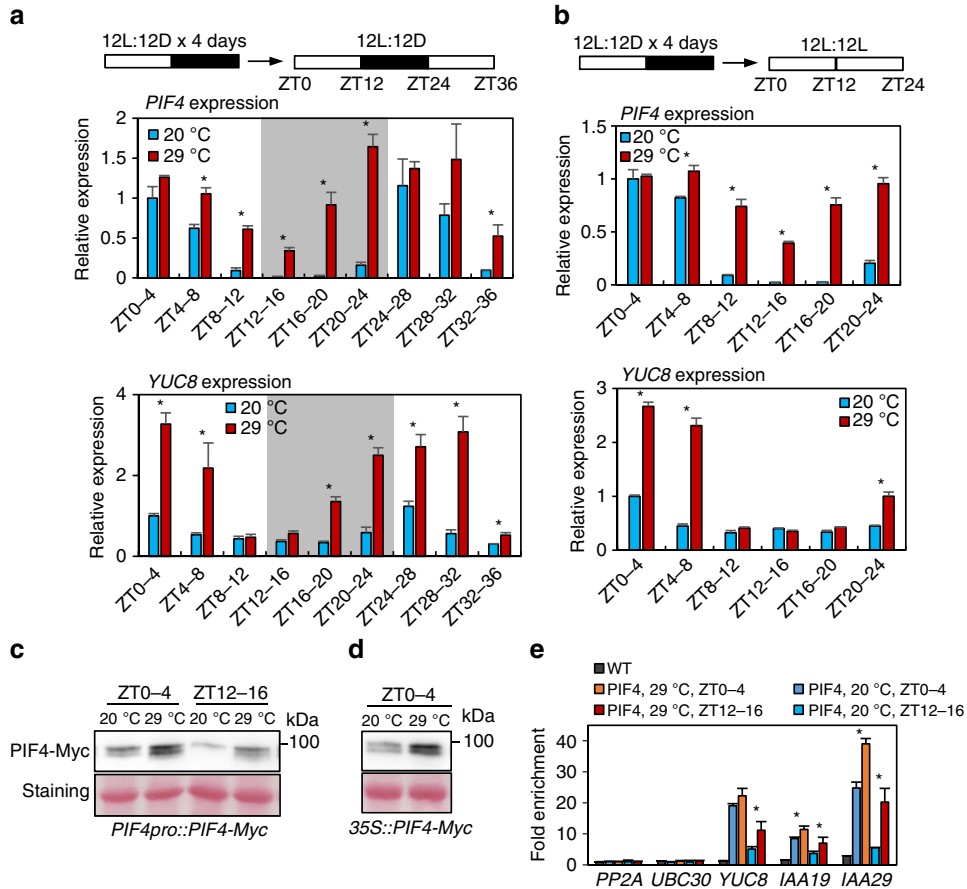

**Figure 4 | The circadian clock gates thermomorphogenesis.** (**a**) Effects of warm temperature at different ZTs on the expression of *PIF4* and *YUC8*. Wild-type seedlings were grown in 12 h light/12 h dark cycles (12L:12D) at 20 °C for 4 days. On the 5th day, the seedlings were treated with warm temperature (29 °C) for 4 h at different ZT (ZT0–ZT36) before harvesting for RNA extraction. At different ZTs, the growth temperature was increased to 29 °C or kept at 20 °C for 4 h. Gene expression levels were normalized to *PP2A* and presented as values relative to that of wild type at ZT0. Error bars indicate s.d. (*n* = 3). *P < 0.05 (Student's *t*-test). (**b**) Effects of warm temperature at different ZTs on the expression of *PIF4* and *YUC8*. Wild-type seedlings were entrained in 12L:12D cycles at 20 °C for 4 days and then transferred under the continuous light. At different ZTs, the growth temperature was increased to 29 °C or kept at 20 °C for 4 h. Gene expression levels were normalized to *PP2A* and presented as values relative to that of wild type at ZT0. Error bars indicate s.d. (*n* = 3). *P < 0.05 (Student's *t*-test). (**c,d**) Warm-temperature effects on PIF4 protein levels. *PIF4p::PIF4-Myc* seedlings (**c**) or *35S::PIF4-Myc* seedlings (**d**) were grown in the same condition as (**b**). Immunoblotting was probed using an anti-Myc antibody and stained with Ponceau S. The experiments were repeated with similar results. The molecular weight (kDa) is indicated on the right side of the gel. (**e**) ChIP-quantitative PCR assays of PIF4-Myc binding to *YUC8, IAA19* and *IAA29* promoters. The *PIF4p::PIF4-Myc* seedlings were grown in the same condition as (**d**). The enrichment of DNA was normalized to that of the *PP2A* coding region as an internal control. Error bars indicate s.d. (*n* = 3). *P < 0.05 (Student's *t*-test).

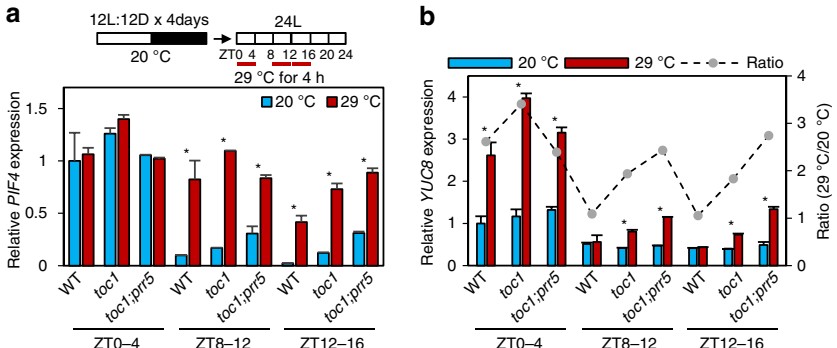

**Figure 5 | TOC1 and PRR5 mediate the circadian gating of thermomorphogenesis.** (**a,b**) Effects of warm temperature at different ZTs on the expression of *PIF4* (**a**) and *YUC8* (**b**). Seedlings were entrained in 12 h light and 12 h dark (12L:12D) cycles at 20 °C for 4 days and then transferred to continuous light. At different ZTs, the growth temperature was increased to 29 °C or kept at 20 °C for 4 h. Seedlings were then harvested at ZT4, ZT16 and ZT20 for RNA extraction. Gene expression levels quantified by qRT–PCR were normalized to *PP2A* and to that of wild type at ZT0 and 20 °C. Grey dots in **b** indicate the ratio of *YUC8* expression levels between 29 °C and 20 °C samples. Error bars indicate s.d. (*n* = 3). *P < 0.05 (Student's *t*-test).

or ZT12–16, *YUC8* expression did not show obvious response (Fig. 5a,b). However, although the basal expression levels of *YUC8* were still low in the evening compared with those in the morning, in both *toc1* and *toc1;prr5* double mutant, the response of *YUC8* expression to warm temperature was partially restored in the *toc1* mutant and more restored in the *toc1;prr5* mutant (Fig. 5b), indicating that the gating of temperature response of *YUC8* requires TOC1 and PRR5. The warm-temperature sensitivity of *YUC8* expression at night of normal light/dark cycles was also restored in the *toc1;prr5* mutant (Supplementary Fig. 8). These results demonstrate that TOC1 and PRR5 mediate the circadian gating of PIF4 transcription activation activity and thermoresponsive growth.

**PIF4 is required for thermo-adaptation.** Previous studies have suggested that plant architectural adaptation to warm tempera-ture, mediated by PIF4, enhances evaporative leaf cooling and thus may improve plant survival under heat stress[4,7]. However, the role of PIF4 in adaptive survival of extreme warm temperature has not been tested. We therefore, tested the effect of *pif4* mutation and *PIF4* overexpression (*PIF4-OX*) on plant survival of heat stresses. Wild-type seedlings grown at a warm temperature of 29 °C showed significantly increased tolerance to the 45 °C heat stress compared with those grown at 20 °C, indicating that adaptation at 29 °C improves survival of extreme heat. Such an adaptive effect was much reduced in the *pif4* mutant (Fig. 6a). In addition, the *PIF4-OX* plants grown at 20 °C were more tolerant to the heat stress than wild type

(Fig. 6b). These results suggest that PIF4 is required for thermo-adaptation and overexpression is sufficient to substitute for pre-exposure to moderately elevated temperature. Consistent with TOC1 inhibition of PIF4 activity, *TOC1-OX* plants were less thermotolerant but the *toc1;prr5* double mutant showed more thermotolerance than wild-type plants when grown under warm temperature (Fig. 6c). When grown under normal temperature, the *TOC1-OX* and *toc1;prr5* plants showed similar sensitivity to the heat stress (Fig. 6c). These results suggest that adaptive thermotolerance requires PIF4 activation and a low level of TOC1, suggesting that PIF4 activation by daytime warm temperatures, allowed by troughs in the level of TOC1, may enhance plant survival during heat stress.

## Discussion

The circadian clock matches biological activities with daily environmental changes associated with the day/night cycles and seasonal changes as well. The circadian gating of the adaptive responses to environmental stresses optimizes survival, while minimizing growth and development penalty[28]. Although temperature oscillates with day–night cycles in nature and adaptive growth response to warm temperature is believed to be crucial for surviving heat stresses, how the circadian clock gates the thermoresponsiveness of plant growth has not been fully understood at the physiological or molecular levels, in part because the responsiveness to warm temperature has not been analysed under free-running circadian conditions. In this work, we demonstrated that the circadian clock gates the

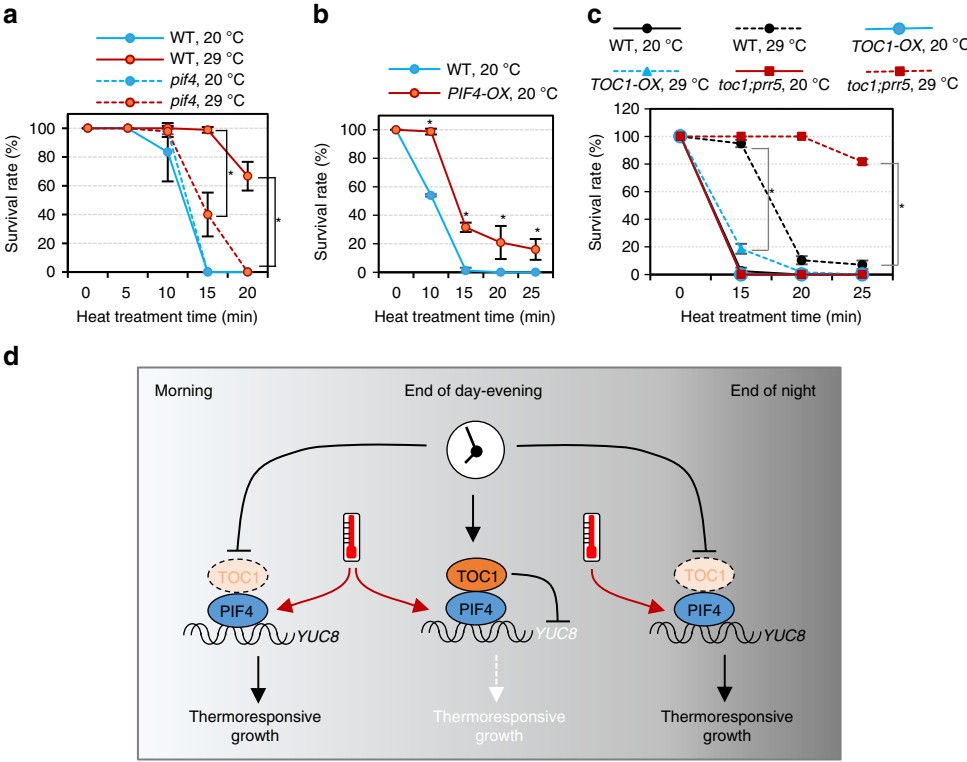

**Figure 6 | PIF4 is required for thermo-adaptation.** (**a,b**) PIF4 is required for thermo-adaptation. Seedlings of WT, *pif4* and *PIF4-OX* genotypes were grown at 20 °C or 29 °C under the continuous light and then were subjected to a 45 °C heat shock for different time periods. After recovery under the constant light at 20 °C for 5 days, survival rate was measured for over 40 seedlings in each sample. The experiments were repeated for three times. Error bars indicate s.d. (*n* = 3). *P < 0.05 (Student's *t*-test). (**c**) The survival rate of WT, *TOC-OX* and *toc1;prr5* seedlings grown at 20 °C or 29 °C, after 45 °C heat-shock treatment for the indicate time periods. Error bars indicate s.d. (*n* = 3). *P < 0.05 (Student's *t*-test). (**d**) A model of the circadian gating of thermomorphogenesis through the TOC1–PIF4 interaction. During the day, warm temperature activates PIF4, which in turn activates auxin biosynthesis genes including *YUC8* and promotes hypocotyl growth. However, in the evening and at early night, TOC1 accumulates at high levels and directly inhibits PIF4, suppressing thermomorphogenesis.

morphological responses to warm temperature through direct inactivation of the thermoresponsive factor PIF4 by the clock component TOC1 in the evening (Fig. 6d). Consistent with TOC1 peaking in the evening, plants are least responsive to warm temperature in the evening. Although the PIF4 level is increased by warming temperature throughout the circadian cycle, maximum activation of PIF4 target genes was observed at the end of night and during morning when both TOC1 and PRR5 levels are low. Around evening time, accumulated TOC1 and PRR5 inhibit PIF4's ability to activate target gene transcription and thus the warm temperature is unable to promote thermoresponsive growth (Fig. 6d). As such, the circadian gating provides maximum thermoresponsiveness to the end of night and early half of the day when temperature increases in nature.

As a central hub for growth regulation, PIF4 is regulated by multiple signals in complex manners. In addition to temperature and the circadian clock, light and hormones also regulate the level of PIF4 activity[8,9,12,13,29,30]. Therefore, thermoresponsiveness is not only controlled by the clock, but also influenced by light conditions. Both phytochromes and cryptochromes directly inhibit PIFs[5,12,13]. In addition to direct effects on PIFs, photoperiod also influences the level of TOC1, which reaches higher level under long-day than short-day conditions[31]. Owing to such complex regulation of PIF4, plants may display different diurnal rhythms of thermoresponse under different light intensities and photoperiods[17,18]. By analysing the responses to short, warm-temperature treatment in plants entrained in light–dark cycles and then shifted into constant light, our study reveals a specific effect of the circadian clock on thermosensitive growth, which is consistent with the evening-specific TOC1-mediated inhibition of PIF4 (Fig. 6d). On the other hand, the TOC1 inhibition of PIF4 is likely to also contribute to clock gating of other PIF-mediated processes such as plant responses to light signals[32]. During the submission of our work for publication, Soy et al.[32] reported that TOC1 mediated the rhythmic hypocotyl growth in short-day condition by temporally constraining the growth-promoting activity of the accumulating PIF3 at the beginning of the long night.

Previous studies have shown a major role of the EC including ELF3 in repressing PIF4 transcript level and circadian inhibition of hypocotyl growth in the early evening[26]. Warm temperature induces PIF4 expression by inhibiting ELF3 binding to PIF4 promoter[17,18]. Such ELF3-mediated thermoregulation of PIF4 expression has been shown to contribute to thermomorphogenesis[17,18]. Consistent with these previous studies, we show that PIF4 RNA level decreases during evening when ELF3 level peaks, but is recovered by a 4 h-warm-temperature treatment. However, the warm-temperature-induced increase of PIF4 level is unable to increase PIF4 target gene expression in the evening, due to TOC1-mediated inactivation of PIF4. As the circadian clock reduces expression of TOC1 at the end of night, the warm-temperature-activated PIF4 is able to induce target gene expression at the end of night and early morning. Such TOC1-mediated circadian gating is consistent with previous observations that the thermosensitivities of PIF4 target gene expression and hypocotyl growth are increased at the end of night under short-day conditions[17,18]. Thus, although the combined actions of ELF3 and TOC1 appear to gate the PIF4-mediated growth, they seem to play different roles in mediating or blocking warm-temperature regulation of PIF4.

A recent proteomic study found that ELF3 directly interacts with TOC1 (ref. 33). In addition, in the PIF4 promoter, the binding motif of TOC1, EE-like-expanded motif, is close to the EC binding sequence, LUX-binding site-like sequence

(GATWCK; Supplementary Fig. 7)[26]. Thus, it seems likely to be that ELF3 and TOC1 may cooperatively repress PIF4 transcription. Further, as ELF3 has been shown to interact directly with PIF4 and prevent PIF4 from binding target DNA[24], it is also possible that ELF3 and TOC1 cooperatively repress PIF4 protein activity[33]. However, unlike ELF3, TOC1 did not interfere with PIF4 DNA-binding ability (Fig. 2c) and TOC1-OX suppressed the thermomorphogenesis and expression of PIF4 target genes in the elf3-null mutant background (Fig. 3k,l and Supplementary Fig. 3), suggesting that TOC1 can repress PIF4 activity in the absence of ELF3. Although both ELF3 and TOC1 repress PIF4 activity and growth in the evening, warmth apparently relieves the ELF3-mediated suppression of PIF4 expression but not the TOC1-mediated posttranslational gating of PIF4 activity. TOC1 expression level is also not obviously affected by warm temperature (Supplementary Fig. 9). Therefore, TOC1 plays a major role in restricting thermoresponsive growth to the end of night and early half of day.

Responses to temperature at different times of the day may serve different physiological functions. Although night temperature can affect plant growth and metabolism, influencing phenotypes such as flowering and fruit development[34,35], adaptive responses to day temperature could be important for survival of heat stress, as the hottest temperature usually occurs around noon to early afternoon and heat is more detrimental to plants under light than in the dark due to photo-oxidative damage[36]. Thus, the TOC1-mediated timing of PIF4 activation by daytime warm temperature could potentially be important for thermo-adaptation and survival of heat stresses in nature. Indeed, pretreatment of wild-type seedlings at warm temperature (29 °C) enhances survival of heat shock (45 °C) treatment (Fig. 6a). Such adaptation is largely lost in the pif4 mutant and TOC1-OX plants (Fig. 6a), confirming the requirement of PIF4 and low TOC1 levels for thermo-adaptation, which occurs during the daytime in nature. It is likely to be that both morphological changes, leading to enhanced cooling capacity[4], and the repression of photosynthetic genes by PIF4 (ref. 9), leading to reduced photo-oxidative damage, contribute to survival under heat stresses. Our study suggests that the circadian gating mediated by the TOC1–PIF4 interaction may enable plants to temporally match adaptive thermo-responses with stress impact during natural day/night cycles to maximize survival of heat stresses.

## Methods

**Plant materials and growth conditions.** A. thaliana plants were grown in a greenhouse with light/dark (16L/8D) cycles at 22 °C for general growth and seed harvesting. All the A. thaliana plants used in this study were in Col-0 ecotype background. The TOC1-OX, toc1-2 and toc1;prr5 mutant seeds were kindly provided by Yamashino et al.[14,37]. The TOC1p::TOC1-YFP seeds were provided by Kay and colleagues[22]. The elf3-1 mutant seeds were provided by Elaine M. Tobin[38]. The PRR5-OX seeds were provided by Nakamichi et al.[20]

**Hypocotyl length measurements.** Sterilized seeds by 70% (v/v) ethanol and 0.01% (v/v) Triton X-100 were plated on Muragshige and Skoog (MS) medium (PhytoTechnology Laboratories) supplemented with 0.75% phytoagar. After 3 days of incubation at 4 °C, seeds were irradiated by white light for 6 h to promote germination and then incubated in specific light conditions (light intensity: 40 $\mu$mol m$^{-2}$ s$^{-1}$). Seedlings were photocopied and hypocotyl lengths were measured by using ImageJ software (http://rsb.info.nih.gov/ij).

**Co-IP assays.** For co-IP assays using Arabidopsis mesophyll protoplasts, $2 \times 10^4$ isolated mesophyll protoplast were transfected with a total of 20 $\mu$g of DNA (35S::TOC1-Myc and 35S::PIF4-YFP) and incubated overnight. Total proteins were extracted from the transfected protoplasts using the IP buffer (50 mM Tris-Cl pH 7.5, 1 mM EDTA, 75 mM NaCl, 0.1% Triton X-100, 5% Glycerol, 1 mM phenylmethylsulfonyl fluoride, 1 × Protease Inhibitor). After centrifugation at 20,000 g for 10 min, the supernatant was incubated for 1 h with anti-green fluorescent protein (GFP; 5 $\mu$g, home-made) immobilized on protein A/G agarose beads (Pierce Biotechnology). The beads were then washed for three times with the IP

buffer and eluted samples were analysed by immunoblotting using anti-Myc (1:5,000 dilution, Cell Signaling, 2276S) and anti-GFP antibodies (1:5,000 dilution, Clontech, 632381). The original gel images are shown in Supplementary Fig. 10.

**Yeast two-hybrid assays.** To detect PIF4 interactions with TOC1/PRRs, the various fragments of TOC1/PRRs or PIF4 complementary DNA were subcloned into the gateway compatible pGADT7 or pGBKT7 vector (Clontech). The resulting yeast constructs containing various fragments of TOC1/PRRs or PIF4 cDNA were co-transformed into yeast AH109 cells (Clontech). The yeast clones were grown on the synthetic dropout medium with histidine ( + His) or without histidine ( − His) containing 1 or 5 mM 3-amino-1, 2, 4-triazole.

**Transient gene expression assays.** Isolated *Arabidopsis* mesophyll protoplasts ($2 \times 10^4$) were transfected with a total 20 µg of DNA and incubated overnight. Protoplasts were harvested by centrifugation and lysed in the passive lysis buffer (Promega). Firefly and *Renilla* (as an internal standard) luciferase activities were measured by using a dual-luciferase reporter kit (Promega). GUS activities were determined by fluorometry with 4-methyl-umbelliferyl glucuronide as substrate.

**Western blot analyses.** Five-day-old seedlings grown at different temperature were harvested and ground in liquid nitrogen. Proteins were extracted with protein extraction buffer (100 mM Tris-HCl pH 6.8, 25% glycerol, 2% SDS, 0.01% bromphenol blue and 10% β-mercaptoethanol). PIF4-Myc protein levels were determined by western blottings using anti-Myc antibody (1:5,000 dilution, Cell Signaling, 2276S). The original gel images are shown in Supplementary Fig. 10.

**Quantitative real-time PCR gene expression analysis.** Total RNA was extracted from seedlings by using the Spectrum Plant Total RNA kit (Sigma). M-MLV reverse transcriptase (Thermal Fisher) was used to synthesize cDNA from the RNA. Quantitative real-time PCR was performed using LightCycler 480 (Roche) and the Bioline SYBR green master mix (Bioline). Gene expression levels were normalized to that of *PP2A* and are shown relative to the expression levels in wild type. Gene-specific primers are listed in Supplementary Table 1.

**ChIP assays.** The wild-type, *PIF4p::PIF4-Myc* or *TOC1p::TOC1-YFP* seedlings were cross-linked for 20 min in 1% formaldehyde solution under vacuum. The cross-linked chromatin complex was isolated by nuclear lysis buffer (50 mM HEPES at pH 7.5, 150 mM NaCl, 1 mM EDTA, 1% Triton X-100, 0.1% Na deoxycholate and 0.1% SDS) and sheared by sonication to reduce the average DNA fragment size to around 500 bp. The sonicated chromatin complex was then immunoprecipitated by anti-Myc antibody for PIF4 ChIP (2 µl, Myc-Tag (9B11) mouse monoclonal antibidy; Cell Signaling) or anti-GFP antibody for TOC1 ChIP (10 µg; home-made). The beads were washed with low-salt buffer (50 mM Tris-HCl at pH 8.0, 2 mM EDTA, 150 mM NaCl and 0.5% Triton X-100), high-salt buffer (50 mM Tris-HCl at pH 8.0, 2 mM EDTA, 500 mM NaCl and 0.5% Triton X-100), LiCl buffer (10 mM Tris-HCl at pH 8.0, 1 mM EDTA, 0.25 M LiCl, 0.5% NP-40 and 0.5% deoxycholate) and TE buffer (10 mM Tris-HCl at pH 8.0 and 1 mM EDTA), and eluted with elution buffer (1% SDS and 0.1 M NaHCO₃). After reverse cross-linking, the DNA was purified with a PCR purification kit (Thermol Fisher) and analysed by ChIP–quantitative PCR. Enrichment was calculated as the ratio between transgenic (*PIF4p::PIF4-Myc* or *TOC1p::TOC1-YFP*) and wild-type samples, normalized to that of the *PP2A*. Primers for the ChIP–quantitative PCR are listed in Supplementary Table 1.

**Heat-shock treatment assays.** For heat-shock treatment, sealed plates containing 7-day-old seedlings were immersed in water bath (45 °C) for different time periods (0–30 min). After the heat-shock treatment, seedlings were allowed to recover at 20 °C. After 5 days, seedling survival rate was scored.

**ChIP-seq analyses.** Previous PIF4, TOC1 and PRR5 ChIP-Seq data were used for the comparative analyses of PIF4 and TOC1 target genes[9,19,20]. Overlaps between PIF4 target genes and TOC1 or PRR5 target genes were analysed by Fisher's exact one-tailed test in R (http://www.r-project.org/). To identify TOC1-binding motifs of TOC1 and PIF4 shared target genes, the sequences of TOC1-binding sites in the shared target genes of TOC1 and PIF4 were analysed by RSAT (Regulatory Sequence Analysis Tools)[39].

**Data availability.** The authors declare that the data supporting the findings of this study are available within the article and its Supplementary Information files or are available from the corresponding author upon request.

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

## Acknowledgements

We thank Takafumi Yamashino for *TOC1-OX*, *toc1-2* and *toc1;prr5* seeds; Steve A. Kay for *TOC1p::TOC1-YFP* seeds; Elaine M. Tobin for *elf3-1* seeds; and Norihito Nakamichi for *PRR5-OX* seeds. Research was supported by a grant from the National Institutes of Health (NIH R01GM066258) and the National Research Foundation of Korea (NRF) grant funded by Korea government (MSIP) (Number 2016R1C1B2008821).

## Author contributions

E.O., J.-Y.Z. and Z.-Y.W. conceived the study and designed the experiments. E.O. and Z.-Y.W. supervised the work. E.O., J.-Y.Z. and T.W. carried out the experiments. E.O., J.-Y.Z. and Z.-Y.W. wrote the manuscript.

## Additional information

**Competing financial interests:** The authors declare no competing financial interests.

