## [Peer Review File · Nature Communications]

Reviewers' comments:

Reviewer #1 (Remarks to the Author):

The paper of Zhu, Oh, Wang and Wang describes the interesting observation that the circadian clock protein TOC1 interacts with PIF4, a central regulator of plant thermomorphogenesis (i.e. growth acclimation in response to a mild increase in temperature). In general, the results seem to support the conclusion that TOC1 interacts with PIF4 and inhibits PIF4 transcriptional activity, thereby suppressing thermomorphogenesis in the evening and early night in a distinct manner from the previously described effect of another circadian regulator; ELF3.

Unfortunately however, the data is often difficult to interpret because figures are ambiguous and minimal information is given in the legends. For instance, different ways of presenting similar data are used (compare e.g. Figure 4a and b). It would aid the paper if the same way of presenting is consequently used (including color codes). In many cases metadata is missing from the main text figure legends (e.g. number of replicates, meaning of error bars and meaning of asterisks, if significance values are present), or the used materials are not explained. What does TOC1:325aa-end for instance mean exactly (legend Sup. Fig 1) and what exactly is shown in e.g. figure 1g and Sup. Fig 1c? The paper would improve from being more consistent.

However, more importantly, proper statistics is missing in most of the figures where key conclusions are drawn from. This includes figure 2e, most panels of Figure 3, figure 4a,b, Figure 4a,b, Figure 5 and figure 6a-c. This lack of statistical comparison makes the interpretation unnecessary ambiguous, especially since the (relative) differences on which conclusions are build are sometime small.

Line 143-149: The TOC1-OX elf3 line has been used in experiments from which it is concluded that "TOC1 inhibits the PIF4 transcriptional activity and thermomorphogenic growth independent of ELF3". An alternative explanation would be that ELF3 operates upstream of TOC1 and that the effect of the elf3 mutation is epistatically masked by TOC1 overexpression. It would be informative to include a elf3 toc1- double mutant in this experiment as well. Would that result in additive stimulated hypocotyl elongation? and toc1-2 ELF3-OX?

At the end of the paper (starting from line 232) the authors used heat-shock experiments to validate the claim that thermomorphogenesis contributes to survival of heat stress. It is not intuitively clear from the start why these distinct experiments are being performed, using a very different temperature and only using the pif4 mutant. Why not toc1-2, TOC1-OX and elf3 as well? Moreover, despite the claim made by the authors, the data to my taste does not support a link between thermomorphogenesis and (line 243): "that the PIF4-mediated thermo-responsive growth enhances plant survival of heat stress". I do agree with the authors that their data show that (line 237): "plants are more sensitive to heat damage during the day time than night". But thermomorphogenesis is a progressive, irreversible response (with the exception of hyponastic growth and not transient and thus I don't think it is likely that dynamic circadian gating of thermomorphogenesis prepares the plant for heat stress over one day, but rather that once induced, it persists over time.

Other comments:

- Are the yeast-two-hybrid experiments replicated (with similar results)? Either mention, or

show the data in the supplements.

- Line 97, figure 2b. Here a 35S::PIF4 line is used to show that PIF4 protein levels are unaltered in TOC1:OX plants. It cannot be ruled out that effects on protein levels are masked by the overexpression of PIF4. Has a immunotagged-PIF4 driven by the endogenous promoter of PIF4 been tested? Also, this and other (Figure 4c, d), protein blot experiments are not described in the materials and methods.
- From Figure 3g and i it appears that PIF4 expression is constitutively lower in TOC1-OX and PRR5-OX lines. Is this effect significant and does this have any biological relevance?
- Explanations of the bioinformatic comparisons of gene expression datasets and the various promoter analyses are lacking from the materials and methods. How are these done? And on what statistical test is the conclusion based that gene expression datasets significantly overlap (Figure 1f and Sup. 1B)?
- At various places in the paper the term 'dramatically' is used to emphasize the outcome of the experiments. Better avoid such ambiguous terms.
- The authors use the term 'heat' throughout the paper when referring to a thermomorphogenesis-inducing mild increase in temperature (from 20oC to 29oC), while in the introduction it is stated that thermomorphogenesis is induced below the heat stress level (line 41). Moreover, 'heat' is also used in the proper context of 'real' heat-treatment (i.e. an increase in temperature up to 45oC). It would help if the use of 'heat' could be restricted to the latter situation.
- Line 47: It is stated that PIF4 interacts with BZR1 and ARF6. Although this is indeed shown in the cited paper, to my knowledge, at least ARF6 has never been shown to contribute to thermomorphogenesis. Therefore, mentioning ARF6 in the context of this paper should perhaps be avoided.
- The authors show that PIF4 protein levels increases under high temperatures (Line 195 and further). This observation was also made before (e.g. by the authors themselves; Oh et al., 2012 and others including (but not exclusively) Foreman et al., 2011). These papers should be referenced.
- Line 201,202: "...significantly increased by warm temperature at all circadian times examined (Fig. 4e)". This is with the exception of YUC8 at ZT0-4.
- Figure 1e; If for the input control Anti-GFP has been used, why then there is a band visible at the PIF4-GFP - (minus) lane? Should this be Anti-Myc instead?
- Although generally well written, at some instances the use of the English language could be improved.

Reviewer #2 (Remarks to the Author):

A) Summary of the key results

The work by Jia-Ying Zhu, Eunkyoo Oh, Tina Wang and Zhi-Yong Wang shows that the induction of YUC8 gene expression in response to warm temperature is gated by the clock. Previous studies had demonstrated that warm temperatures increase the expression of PIF4, which in turn binds and enhances the activity of the YUC8 promoter. However, the

gating of the temperature response reported here is not accounted for by the daily patterns of PIF4 gene expression. The time of low sensitivity to warm temperature (evening, early night) overlaps with the time of expression of the clock gene TOC1. The authors confirm and extend previous reports supporting physical interaction between TOC1 and PIF4 and demonstrate that TOC1 acts as repressor of PIF4 transcriptional activity. TOC1 reduces the growth response to warm temperature in a PIF4-dependent manner. The *toc1* mutation allows YUC8 expression responses to temperature during the evening and early night.

B) Originality and interest

The gating of the temperature response reported here is of strong biological significance because temperature oscillates during day/night cycles. Restricting the responses to a given portion of the day would buffer the impact of diurnal fluctuations. The study presents a careful analysis of the molecular mechanism mediating this control. Therefore, the work provides significant insight into our understanding of the control of plant growth by temperature.

C) Conclusions: robustness, validity, reliability

The conclusions of the manuscript are well supported by the data. The interpretation of one of the experiments does not appear to be correct but the experiment can be excluded without affecting the significance of the work.

The results presented in Fig. 6a-c are interesting. However, they are not connected to the rest of the work. All the manuscript deals with growth responses to temperature whereas in these figures the response is seedling survival. The authors link both responses (e.g. "PIF4-mediated thermo-adaptation enhances plant survival of heat stresses") but the explanation is not convincing. For instance, in Fig. 6a, the seedlings have better survival at ZT12 h, when PIF4 activity and the growth response to temperature are restricted by TOC1. Furthermore, the exposure to elevated temperatures is only for 16 min, which is adequate for the experiment, but clearly not enough to modify the growth pattern of the plant to alleviate the impact of the ongoing stress. The growth response to warm temperatures and the acquisition of tolerance to heat shocks are different processes; although growth patterns can potentially affect seedling survival they require time to have a real impact. These data should be removed to avoid misunderstandings.

D) Data & methodology: validity of approach, quality of data, quality of presentation

The methodology is correct and the results show high quality and completeness.

E) Appropriate use of statistics and treatment of uncertainties

1) Fig 1 (f) Please provide statistical test to indicate the significance of the overlap.

2) Fig. 1 (i) The analysis should include PIF4 only to indicate whether the feature is either dependent on the combination of TOC1 and PIF4 or only on PIF4.

3) Fig. 4 (c, d) describes warm temperature effects on PIF4 protein levels. It would be ideal to quantify the bands from different biological samples and provide statistical significance. At least, indicate the number of biological samples that have shown the same pattern.

F) Clarity and context:

The manuscript is well written but the following issues should be addressed:

1) Figure 3. Some figures change the format for similar experiments. It would be helpful to keep consistent use of colours for the temperature treatments.

2) According to the legend, Fig. 3(k) shows hypocotyl lengths of TOC1-OX;elf3-1 seedlings grown as described in (h, i). There must be something wrong because the large differences in hypocotyl length reported in this figure cannot be generated in 4 h.

3) Fig 1 (e) (Co-immunoprecipitation assays). Is labelling correct?

G) Suggested improvements: experiments, data for possible revision

The authors argue that "While both ELF3 and TOC1 repress PIF4 activity and growth in the evening, warmth apparently relieves the ELF3-mediated suppression of PIF4 expression but not the TOC1-mediated gating of PIF4 activity". Temperature enhances the expression of the PIF4 gene and the stability of the PIF4 protein. Any information about temperature effects on TOC1 gene expression and/or protein stability?

H) References.

References are adequate.

Reviewer #3 (Remarks to the Author):

In this manuscript, the authors investigate whether binding of TOC1 to PIF4 in the evening serves to repress the thermomorphogenesis response promoted by PIF4. The authors employ a mix of in vitro, genomic and molecular techniques to show PIF4 and TOC1 physically interact and these transcription factors have substantial overlap in target genes. They also show TOC1 association with PIF4 in vitro inhibits transcriptional activation by PIF4. Genetic and gene expression tests show the toc1-2 mutant and TOC1 overexpression enhance and repress, respectively, PIF4-dependent thermoresponsive hypocotyl. Similar results are shown for overexpression of PRR5-OX. ChIP of TOC1 shows it is associated with established PIF4 target promoters. To evaluate a role for the TOC1-PIF4 interaction in gating of thermomorphogenesis, several sets of are presented that correlate evening inhibition of PIF4 activity and TOC1 activity. Finally, the authors test the functional link between PIF4 activity gating and heat shock survival.

The manuscript is data rich and the experiments are thoughtfully presented. All of the data support the their model, with the exception of the tenuous link between heat shock survival and the gating model. While survival of heat shock appears time-of-day dependent, with heat being more lethal in at dawn, it is difficult to rationalize how PIF4 promotes heat stress survival given that the thrust of this manuscript is that TOC1 inhibits PIF4 activity at dusk.

Figures 2b and 4c,d, show Ponceau staining of membranes to demonstrate equal loading/transfer for these Western blots. This technique is neither sensitive nor quantitative. Probing for a constitutive protein like tubulin is necessary to confirm equivalent amounts of protein per lane.

The text needs some attention to improve English use throughout.

Minor issues

Line 110 and several other places . The authors use the term "transcriptional activity" to describe the function of PIF4. This term is not accurate. RNA polymerase has transcriptional activity. PIF4 either activates or represses transcription.

Lines 159-161. Thines et al. (2010 PNAS) previously described the identical gating behavior for PIF4 expression. This paper should be cited.

Lines 526-538. The authors need to clarify the legend for Figure 4. It is not clear what conditions were employed here. Also, there is reference to panel f, but the figure lacks this panel.

Line 280, replace "cryptocheomes" with "cryptochromes".

Reviewer #4 (Remarks to the Author):

The plant circadian clock has been shown to regulate the expression of genes involved in temperature responses but the physiological significance of this regulation is not known. This paper shows that the circadian clock acts to gate thermogenic responses (hypocotyl elongation and expression of the genes YUC8, IAA19 and IAA29) to the late night and early day. These responses were previously shown to be controlled by the PIF4 transcription factor. Warm temperatures inhibit binding of the circadian clock component ELF3 to the PIF4 promoter, leading to its increased expression. However, the authors now show that the evening-expressed proteins TOC1 and PRR5 bind the PIF4 protein and suppress its transcriptional activity. Plants exposed to warm temperatures in the evening exhibit increased levels of PIF4 protein, but expression of PIF4 target genes is not induced due to its inactivation by PRR5 and TOC1. Consequently plants don't alter their morphology in response to warm temperatures in the evening. Thus the circadian clock acts to ensure that plants only alter their morphology in response to warm temperatures in the morning, in anticipation of hot temperatures during the day, but ignore warm temperatures in the

evening when cooler temperatures are expected at night.

These findings are novel and exciting and will be of interest to the broad readership of Nature Communications. The paper is clearly written (although it may need a few small editorial changes), experiments are elegant and thorough and the quality of the data is excellent.

Nevertheless, I have reservations regarding the findings in Figure 6, which the authors will need to address: Figure 6 shows that plants are increased tolerance to heat stress in the evening than at night, that plants acclimated to warm temperature have increased heat tolerance, that this requires PIF4 function, and that increased PIF4 expression results in improved tolerance to heat stress.

However while the results appear robust it is unclear whether these findings are related to the gating of thermo-responsive growth mediated by the TOC1-PIF4 interaction, because maximum tolerance to heat stress is observed at the time of the day when TOC1 suppresses responses to high temperature. Furthermore, morphological changes and reduction in expression of photosynthetic genes may play a role in adaptation to warm temperatures but are unlikely to take place over the short duration of a heat shock (15-20 minutes).

Therefore the mechanisms proposed in lines 316-318 are unrealistic.

While Figure 6b and 6c shows that PIF4 contributes to adaptation to elevated temperatures, they don't show whether it contributes to rhythmic changes in heat shock tolerance. It is also unclear whether TOC1 and PRR5 play a role. In order to address these issues, the authors need to repeat the experiments in Figure 6a with *toc1*; *prp5* and *pif4* mutants. If what they claim is correct, all of these mutants should show similar sensitivity to heat shock at ZT0 and at ZT12.

An alternative hypothesis is that the rhythmic changes in heat shock tolerance are due to ELF3-mediated regulation of PIF4 transcription. Therefore the experiments in Figure 6a also need to be repeated with the *elf3* mutant. If the rhythmic changes in heat shock tolerance are regulated by ELF3 and PIF4 but not by TOC1 this would suggest that there is a subset of PIF4 targets that is not regulated by TOC1 and that is responsible for increased heat shock tolerance in the evening.

Minor comments:

Page 8, line 1. ZT needs to be defined (Zeitgeber time, i.e. hours after dawn).

Page 11, line 224: "the response of YUC8 was partially restored in the *toc1* mutant and more restored in the *toc1*; *prp5* mutant". Here the raw data are somewhat misleading as the YUC8 expression levels remain relatively low in the double mutant in response to warm temperature in the evening as compared to morning. However the fold-induction in the double mutant is comparable in the day time and at night, showing that the gating of the response is abolished. The authors should explain this more clearly.

Suggested changes to figure legends:

Lines 480-481: Yeast clones were grown on synthetic dropout medium plus 1 mM 3AT, with

or without histidine (+ HIS or -HIS)

Lines 485-486: Overlap between PIF4 and Toc1 target genes identified in ChIP-seq assays.

Lines 493-493. IAA19p::luc was co-transfected with PIF4-GFP, TOC1-GFP and 35S:: renilla luciferase into Arabidopsis mesophyll chloroplasts. Luciferase activity levels were normalized to Renilla luciferase activity.

Line 500: then used for ChIp assays using a MYC antibody.

Line 502: the UAS-GUS reporter construct was co-transfected...

Line 547: ZTs

Figure 3: it would make sense to use similar graphs and color schemes for similar assays. Why the different format for panel g?

Figure 4 panel E should indicate PIF4-Myc in the Figure legend rather than just PIF4. It would make more sense when interpreting the data.

Reviewer #1 (Remarks to the Author):

The paper of Zhu, Oh, Wang and Wang describes the interesting observation that the circadian clock protein TOC1 interacts with PIF4, a central regulator of plant thermomorphogenesis (i.e. growth acclimation in response to a mild increase in temperature). In general, the results seem to support the conclusion that TOC1 interacts with PIF4 and inhibits PIF4 transcriptional activity, thereby suppressing thermomorphogenesis in the evening and early night in a distinct manner from the previous described effect of another circadian regulator; ELF3.

- Unfortunately however, the data is often difficult to interpret because figures are ambiguous and minimal information is given in the legends. For instance, different ways of presenting similar data are used (compare e.g. Figure 4a and b). It would aid the paper if the same way of presenting is consequently used (including color codes). In many cases metadata is missing from the main text figure legends (e.g. number of replicates, meaning of error bars and meaning of asterisks, if significance values are present), or the used materials are not explained. What does TOC1:325aa-end for instance mean exactly (legend Sup. Fig 1) and what exactly is shown in e.g. figure 1g and Sup. Fig 1c? The paper would improve from being more consistent. However, more importantly, proper statistics is missing in most of the figures where key conclusions are drawn from. This includes figure 2e, most panels of Figure 3, figure 4a,b, Figure 4a,b, Figure 5 and figure 6a-c. This lack of statistical comparison makes the interpretation unnecessary ambiguous, especially since the (relative) differences on which conclusions are build are sometime small.

Response: We have provided these missing details in our revised manuscript. In the texts (page 5), we pointed out that Figure 1g and Sup. Fig 1c have shown ‘that the binding sites of PIF4 and TOC1 in the shared target genes were close to each other, suggesting that TOC1 and PIF4 tend to bind to the same genomic locations.’

- Line 143-149: The TOC1-OX *elf3* line has been used in experiments from which it is concluded that "TOC1 inhibits the PIF4 transcriptional activity and thermomorphogenic growth independent of ELF3". An alternative explanation would be that ELF3 operates upstream of TOC1 and that the effect of the *elf3* mutation is epistatically masked by TOC1 overexpression. It would be informative to include a *elf3 toc1*- double mutant in this experiment as well. Would that result in additive stimulated hypocotyl elongation? and *toc1-2 ELF3-OX*.

Response: The result supports the conclusion that, at the biochemical level, TOC1 is able to inhibit PIF4 activity in the absence of ELF3. We have changed the sentence to “TOC1 is able to inhibit the PIF4 activity and thermomorphogenic growth in the absence of ELF3.”. This conclusion does not conflict with the reviewer’s scenario that ELF3 acts upstream of TOC1. In fact, it has been shown that TOC1 expression level is reduced in the *elf3* mutant (Thines and Harmonm, 2010), and thus we do not need to retest whether ELF3 operates upstream of TOC1. Instead, our question is specifically whether ELF3, which interacts with PIF4, is required for TOC1 protein to repress PIF4 activity.

- At the end of the paper (starting from line 232) the authors used heat-shock experiments to validate the claim that thermomorphogenesis contributes to survival

of heat stress. It is not intuitively clear from the start why these distinct experiments are being performed, using a very different temperature and only using the *pif4* mutant. Why not *toc1-2*, *TOC1-OX* and *elf3* as well? Moreover, despite the claim made by the authors, the data to my taste does not support a link between thermomorphogenesis and (line 243): "that the PIF4-mediated thermo-responsive growth enhances plant survival of heat stress". I do agree with the authors that their data show that (line 237): "plants are more sensitive to heat damage during the day time than night". But thermomorphogenesis is a progressive, irreversible response (with the exception of hyponastic growth and not transient and thus I don't think it is likely that dynamic circadian gating of thermomorphogenesis prepares the plant for heat stress over one day, but rather that once induced, it persists over time.

Response: We thought that analyzing *toc1-2*, *TOC1-OX* and *elf3* for thermo-tolerance is unnecessary because PIF4 is the key factor controlling thermomorphogenesis. While TOC1 and ELF4 regulate PIF4 level/activity, they regulate not only PIF4 but also other factors and thus their effects may be complex and not specific to PIF4 and thermo responses. However, we have done similar experiments with *TOC1-OX* and *toc1;prp5* double mutants. Consistent with the TOC1 inhibition of thermomorphogenesis, *TOC1-OX* grown at 29°C was more susceptible to heat shock, *toc1;prp5* double mutants grown at 29°C were more resistant to the heat shock than the wild type plants. We have added these data in Supplementary Fig. 9.

We think the reviewer misunderstood our conclusion. We also do not believe that the circadian gating prepares plants for heat stress over one day. Instead, the circadian gating allows the plants to grow according to the temperature of the day, not of the night, and therefore adapt a morphology that enhances survival of further increase of day-time temperature. We have changed our discussion to clarify this point. The link between thermomorphogenesis and heat stress tolerance has been suggested in the previous literature (e.g. Crawford et al., 2012 Curr Bio), but has only been demonstrated experimentally for the first time by our study.

Other comments:

- Are the yeast-two-hybrid experiments replicated (with similar results)? Either mention, or show the data in the supplements.

Response: We have tested at least three independent replicates for yeast two hybrid assays. We have added the results of all of the yeast clones in Supplementary Fig. 1.

- Line 97, figure 2b. Here a 35S::PIF4 line is used to show that PIF4 protein levels are unaltered in *TOC1:OX* plants. It cannot be ruled out that effects on protein levels are masked by the overexpression of PIF4. Has a immunotagged-PIF4 driven by the endogenous promoter of PIF4 been tested? Also, this and other (Figure 4c, d), protein blot experiments are not described in the materials and methods.

Response: In figure 2b, to test if TOC1 directly affects PIF4 protein stability, we have checked 35S promoter-driven PIF4 protein instead of *PIF4* native promoter-driven PIF4 protein because TOC1 (together with PRR5) also affects *PIF4* RNA expression (Figure 5). The point of figure 2b is that the TOC1 inhibition of PIF4 transcriptional activity observed in figure 2a (also used 35S::PIF4) is not due to an altered PIF4 protein level by TOC1.

We have added the experimental descriptions for figure 2b, 4c and 4d in Methods.

- From Figure 3g and it appears that PIF4 expression is constitutively lower in TOC1-OX and PRR5-OX lines. Is this effect significant and does this have any biological relevance?

Response: As we have shown in Fig. 5 that TOC1 and PRR5 redundantly repress *PIF4* expression at the evening. So, these two transcription factors are also involved in the circadian clock-regulation of *PIF4* expression. Such repression of *PIF4* expression should contribute to low PIF4 activity in the evening/night phase of normal conditions (20~22°C). However, warm temperature (28°C) induces *PIF4* expression but not expression of its target gene around the evening, indicating that posttranslational repression of PIF4 activity by TOC1 and PRR5 suppresses the thermo-response.

- Explanations of the bioinformatic comparisons of gene expression datasets and the various promoter analyses are lacking from the materials and methods. How are these done? And on what statistical test is the conclusion based that gene expression datasets significantly overlap (Figure 1f and Sup. 1B)?

Response: We have added detailed statistical methods for the bioinformatics analysis in Methods.

- At various places in the paper the term 'dramatically' is used to emphasize the outcome of the experiments. Better avoid such ambiguous terms.

Response: We have removed term “dramatically”.

- The authors use the term 'heat' throughout the paper when referring to a thermomorphogenesis-inducing mild increase in temperature (from 20oC to 29oC), while in the introduction it is stated that thermomorphogenesis is induced below the heat stress level (line 41). Moreover, 'heat' is also used in the proper context of 'real' heat-treatment (i.e. an increase in temperature up to 45°C). It would help if the use of 'heat' could be restricted to the latter situation.

Response: We agree and have changed the manuscript as reviewer suggested.

- Line 47: It is stated that PIF4 interacts with BZR1 and ARF6. Although this is indeed shown in the cited paper, to my knowledge, at least ARF6 has never been shown to contribute to thermomorphogenesis. Therefore, mentioning ARF6 in the context of this paper should perhaps be avoided.

Response: We agree and removed ARF6 from the sentence.

- The authors show that PIF4 protein levels increases under high temperatures (Line 195 and further). This observation was also made before (e.g. by the authors themselves; Oh et al., 2012 and others including (but not exclusively) Foreman et al., 2011). These papers should be referenced.

Response: Our previous study have used *PIF4promoter::PIF4*, so it was not clear if high temperature increases PIF4 protein post-transcriptionally as well as transcriptionally. We have added the reference (Foreman et al., 2011).

- Line 201,202: "...significantly increased by warm temperature at all circadian times examined (Fig. 4e)". This is with the exception of YUC8 at ZT0-4.

Response: We have changed the sentence as follows:

'PIF4 binding to its target promoters was significantly increased by warm temperature at all circadian time examined, except for *YUC8* at ZT0-4.'

- Figure 1e; If for the input control Anti-GFP has been used, why then there is a band visible at the PIF4-GFP - (minus) lane? Should this be Anti-Myc instead?

Response: The labeling was wrong. We have corrected Figure 1e.

- Although generally well written, at some instances the use of the English language could be improved.

Response: We have polished the language.

Reviewer #2 (Remarks to the Author):

A) Summary of the key results

The work by Jia-Ying Zhu, Eunkyoo Oh, Tina Wang and Zhi-Yong Wang shows that the induction of YUC8 gene expression in response to warm temperature is gated by the clock. Previous studies had demonstrated that warm temperatures increase the expression of PIF4, which in turn binds and enhances the activity of the YUC8 promoter. However, the gating of the temperature response reported here is not accounted for by the daily patterns of PIF4 gene expression. The time of low sensitivity to warm temperature (evening, early night) overlaps with the time of expression of the clock gene TOC1. The authors confirm and extend previous reports supporting physical interaction between TOC1 and PIF4 and demonstrate that TOC1 acts as repressor of PIF4 transcriptional activity. TOC1 reduces the growth response to warm temperature in a PIF4-dependent manner. The *toc1* mutation allows YUC8 expression responses to temperature during the evening and early night.

B) Originality and interest

The gating of the temperature response reported here is of strong biological significance because temperature oscillates during day/night cycles. Restricting the responses to a given portion of the day would buffer the impact of diurnal fluctuations. The study presents a careful analysis of the molecular mechanism mediating this control. Therefore, the work provides significant insight into our understanding of the control of plant growth by temperature.

C) Conclusions: robustness, validity, reliability

The conclusions of the manuscript are well supported by the data. The interpretation of one of the experiments does not appear to be correct but the experiment can be excluded without affecting the significance of the work.

The results presented in Fig. 6a-c are interesting. However, they are not connected to the rest of the work. All the manuscript deals with growth responses to temperature whereas in these figures the response is seedling survival. The authors link both responses (e.g. "PIF4-mediated thermo-adaptation enhances plant survival of heat stresses") but the explanation is not convincing. For instance, in Fig. 6a, the seedlings have better survival at ZT12 h, when PIF4 activity and the growth response to temperature are restricted by TOC1. Furthermore, the exposure to elevated temperatures is only for 16 min, which is adequate for the experiment, but clearly not enough to modify the growth pattern of the plant to alleviate the impact of the ongoing stress. The growth response to warm temperatures and the acquisition of tolerance to heat shocks are different processes; although growth patterns can potentially affect seedling survival they require time to have a real impact. These data should be removed to avoid misunderstandings.

Response: We can agree to remove these results. However, we think the reviewer misunderstood our experimental design because we did not explain it clearly. Fig. 6a test/confirm previous report that heat stress is enhanced by light, and high temperature does not cause damage if followed immediately by darkness, and therefore, plants should be more sensitive to heat during the day than in the night. There is no morphological effect in this experiment. We have changed our description of Fig 6a. The rest of Fig 6 supports the essential role for PIF4 in heat tolerance.

D) Data & methodology: validity of approach, quality of data, quality of presentation

The methodology is correct and the results show high quality and completeness.

E) Appropriate use of statistics and treatment of uncertainties

1) Fig 1 (f) Please provide statistical test to indicate the significance of the overlap.

Response: According to reviewer's suggestion, we have provided the statistical test.

2) Fig. 1 (i) The analysis should include PIF4 only to indicate whether the feature is either dependent on the combination of TOC1 and PIF4 or only on PIF4.

Response: We have added the PIF4-only data in Fig. 1i.

3) Fig. 4 (c, d) describes warm temperature effects on PIF4 protein levels. It would be ideal to quantify the bands from different biological samples and provide statistical significance. At least, indicate the number of biological samples that have shown the same pattern.

Response: We have repeated this experiment with similar results and indicated this

in figure legends.

F) Clarity and context:

The manuscript is well written but the following issues should be addressed:

1) Figure 3. Some figures change the format for similar experiments. It would be helpful to keep consistent use of colours for the temperature treatments.

Response: We have changed the color of bars in Figure 3c, d, e and i according to reviewer's suggestion.

2) According to the legend, Fig. 3(k) shows hypocotyl lengths of *TOC1-OX;elf3-1* seedlings grown as described in (h, i). There must be something wrong because the large differences in hypocotyl length reported in this figure cannot be generated in 4 h.

Response: We have corrected the legend for Fig. 3k.

3) Fig 1 (e) (Co-immunoprecipitation assays). Is labelling correct?

Response: The labeling was wrong. We have corrected the labelling.

G) Suggested improvements: experiments, data for possible revision

The authors argue that "While both ELF3 and TOC1 repress PIF4 activity and growth in the evening, warmth apparently relieves the ELF3-mediated suppression of PIF4 expression but not the TOC1-mediated gating of PIF4 activity". Temperature enhances the expression of the PIF4 gene and the stability of the PIF4 protein. Any information about temperature effects on TOC1 gene expression and/or protein stability?

Response: *TOC1* gene expression is slightly increased (not decreased) by high temperature treatment (29°C). We have added these results in Supplementary Fig. 10 and Discussion on page 14. In addition, high temperature could not promote hypocotyl elongation in *TOC1-OX* (Fig 3c), consistent with TOC1-mediated suppression being unaffected by temperature.

H) References.

References are adequate.

Reviewer #3 (Remarks to the Author):

In this manuscript, the authors investigate whether binding of TOC1 to PIF4 in the evening serves to repress the thermomorphogenesis response promoted by PIF4. The authors employ a mix of in vitro, genomic and molecular techniques to show

PIF4 and TOC1 physically interact and these transcription factors have substantial overlap in target genes. They also show TOC1 association with PIF4 in vitro inhibits transcriptional activation by PIF4. Genetic and gene expression tests show the *toc1-2* mutant and TOC1 overexpression enhance and repress, respectively, PIF4-dependent thermoresponsive hypocotyl. Similar results are shown for overexpression of PRR5-OX. ChIP of TOC1 shows it is associated with established PIF4 target promoters. To evaluate a role for the TOC1-PIF4 interaction in gating of thermomorphogenesis, several sets of are presented that correlate evening inhibition of PIF4 activity and TOC1 activity. Finally, the authors test the functional link between PIF4 activity gating and heat shock survival.

- The manuscript is data rich and the experiments are thoughtfully presented. All of the data support the their model, with the exception of the tenuous link between heat shock survival and the gating model. While survival of heat shock appears time-of-day dependent, with heat being more lethal in at dawn, it is difficult to rationalize how PIF4 promotes heat stress survival given that the thrust of this manuscript is that TOC1 inhibits PIF4 activity at dusk.

Response: The survival of heat shock is light/dark dependent, as reported previously, and is unlikely time-of-day dependent. We have changes the description of this part (as discussed above in response to reviewer 2).

- Figures 2b and 4c,d, show Ponceau staining of membranes to demonstrate equal loading/transfer for these Western blots. This technique is neither sensitive nor quantitative. Probing for a constitutive protein like tubulin is necessary to confirm equivalent amounts of protein per lane.

Response: In our opinion, staining of the blot is a more reliable visual reference of sample loading for several reasons. First, no “reference protein” is for sure constitutive. In particular, proteins involved in cell elongation such as tubulin and actin are known to be altered by light and hormones. Second, gel transfer can be uneven in different area of the gel, and proteins of different sizes may be affected differently without our knowing it, whereas staining allows evaluation of the whole gel.

- The text needs some attention to improve English use throughout.

Response: We have polished the language.

Minor issues

- Line 110 and several other places. The authors use the term "transcriptional activity" to describe the function of PIF4. This term is not accurate. RNA polymerase has transcriptional activity. PIF4 either activates or represses transcription.

Response: We changed “transcriptional activity” to “ability to activate target gene transcription”.

- Lines 159-161. Thines et al. (2010 PNAS) previously described the identical gating behavior for PIF4 expression. This paper should be cited.

Response: We have added the reference.

- Lines 526-538. The authors need to clarify the legend for Figure 4. It is not clear what conditions were employed here. Also, there is reference to panel f, but the figure lacks this panel.

Response: We have changed the legend for Figure 4 according to reviewer's suggestion and removed the reference to panel f.

- Line 280, replace "cryptocheomes" with "cryptochromes".

Response: We have corrected this typo.

Reviewer #4 (Remarks to the Author):

The plant circadian clock has been shown to regulate the expression of genes involved in temperature responses but the physiological significance of this regulation is not known. This paper shows that the circadian clock acts to gate thermogenic responses (hypocotyl elongation and expression of the genes YUC8, IAA19 and IAA29) to the late night and early day. These responses were previously shown to be controlled by the PIF4 transcription factor. Warm temperatures inhibit binding of the circadian clock component ELF3 to the PIF4 promoter, leading to its increased expression. However, the authors now show that the evening-expressed proteins TOC1 and PRR5 bind the PIF4 protein and suppress its transcriptional activity. Plants exposed to warm temperatures in the evening exhibit increased levels of PIF4 protein, but expression of PIF4 target genes is not induced due to its inactivation by PRR5 and TOC1. Consequently plants don't alter their morphology in response to warm temperatures in the evening. Thus the circadian clock acts to ensure that plants only alter their morphology in response to warm temperatures in the morning, in anticipation of hot temperatures during the day, but ignore warm temperatures in the evening when cooler temperatures are expected at night.

These findings are novel and exiting and will be of interest to the broad readership of Nature Communications. The paper is clearly written (although it may need a few small editorial changes), experiments are elegant and thorough and the quality of the data is excellent.

Nevertheless, I have reservations regarding the findings in Figure 6, which the authors will need to address: Figure 6 shows that plants are increased tolerance to heat stress in the evening than at night, that plants acclimated to warm temperature have increased heat tolerance, that this requires PIF4 function, and that increased PIF4 expression results in improved tolerance to heat stress.

However while the results appear robust it is unclear whether these findings are related to the gating of thermo-responsive growth mediated by the TOC1-PIF4 interaction, because maximum tolerance to heat stress is observed at the time of the day when TOC1 suppresses responses to high temperature. Furthermore, morphological changes and reduction in expression of photosynthetic genes may play a role in adaptation to warm temperatures but are unlikely to take place over the

short duration of a heat shock (15-20 minutes). Therefore the mechanisms proposed in lines 316-318 are unrealistic.

While Figure 6b and 6c shows that PIF4 contributes to adaptation to elevated temperatures, they don't show whether it contributes to rhythmic changes in heat shock tolerance. It is also unclear whether TOC1 and PRR5 play a role. In order to address these issues, the authors need to repeat the experiments in Figure 6a with *toc1*; *prp5* and *pif4* mutants. If what they claim is correct, all of these mutants should show similar sensitivity to heat shock at ZT0 and at ZT12.

An alternative hypothesis is that the rhythmic changes in heat shock tolerance are due to ELF3-mediated regulation of PIF4 transcription. Therefore the experiments in Figure 6a also need to be repeated with the *elf3* mutant. If the rhythmic changes in heat shock tolerance are regulated by ELF3 and PIF4 but not by TOC1 this would suggest that there is a subset of PIF4 targets that is not regulated by TOC1 and that is responsible for increased heat shock tolerance in the evening.

Response: Apparently we failed to explain this part clearly, as several reviewers were confused. Fig 6a tests the effect of light/dark, not of clock, on heat damage, without any adaptation treatment. The results, together with previous reports, demonstrate that heat is more detrimental to plant under light (during day) than in the dark (at night), and therefore adaptive growth response should be based on day-time temperature (as enabled by TOC1 regulation of PIF4) rather than night time temperature (potentially provided by ELF3). Another argument for the same conclusion is that heat stress occurs during summer day time not night. We have cited the previous work showing heat stress is more severe under light than in the dark, and revised our description of this experiment.

Minor comments:

- Page 8, line 1. ZT needs to be defined (Zeitgeber time, i.e. hours after dawn).

Response: According to reviewer's suggestion, we have defined ZT on page 8.

- Page 11, line 224: "the response of YUC8 was partially restored in the *toc1* mutant and more restored in the *toc1*; *prp5* mutant". Here the raw data are somewhat misleading as the YUC8 expression levels remain relatively low in the double mutant in response to warm temperature in the evening as compared to morning. However the fold-induction in the double mutant is comparable in the day time and at night, showing that the gating of the response is abolished. The authors should explain this more clearly.

Response: We have changed the sentence as follows:

'Although the basal expression levels of *YUC8* were still low in the evening compared to those in the morning in both *toc1* and *toc1*; *prp5* double mutant, the response of *YUC8* expression to warm temperature was partially restored in the *toc1* mutant and more restored in the *toc1*; *prp5* mutant, indicating that the gating of *YUC8* response to warmth requires TOC1/PRR5'

- Suggested changes to figure legends:

Lines 480-481: Yeast clones were grown on synthetic dropout medium plus 1 mM

3AT, with or without histidine (+ HIS or -HIS)

Lines 485-486: Overlap between PIF4 and Toc1 target genes identified in ChIP-seq assays.

Lines 493-493. IAA19p::luc was co-transfected with PIF4-GFP, TOC1-GFP and 35S:: renilla luciferase into Arabidopsis mesophyll chloroplasts. Luciferase activity levels were normalized to Renilla luciferase activity.

Line 500: then used for ChIP assays using a MYC antibody.

Line 502: the UAS-GUS reporter construct was co-transfected...

Line 547: ZTs

Figure 3: it would make sense to use similar graphs and color schemes for similar assays. Why the different format for panel g?

Figure 4 panel E should indicate PIF4-Myc in the Figure legend rather than just PIF4. It would make more sense when interpreting the data.

Response: We have made the above suggested changes except for 'Lines 480-481: Yeast clones were grown on synthetic dropout medium plus 1 mM 3-AT, with or without histidine (+ HIS or -HIS)'. According to our experiment procedure, we did not add 1 mM 3-AT in the yeast synthetic dropout medium with histidine, so we think the sentence 'Yeast clones were grown on the synthetic dropout medium (+HIS) or synthetic dropout medium without histidine plus 1 mM 3AT (-HIS).' in figure legend Fig.1(c-d) described the experiment accurately.

REVIEWERS' COMMENTS:

Reviewer #1 (Remarks to the Author):

The revised manuscript of Dr. Wang addresses most issues raised by the reviewers on a reasonable level by adding new data and amending the texts at several places. The authors also significantly improved the texts by explaining the rationale behind testing heat stress tolerance in wild type and pif4 mutant plants pre-subjected to control and warm temperatures (Fig 6). This part of the manuscript was confusing to all four reviewers and although improved, I suggest even more effort need to be taken to make crystal clear what justifies this twist in the flow of the paper. If well described it certainly has the potential to enhance the quality paper, but if not it may appear an odd twist to those readers that are not entirely up to date with the state-of-the-art in the field of thermoresponsive growth. I do agree with the notion of the authors in their response letter that a connection between high ambient pre-growth temperature was suggested before in literature, but never tested experimentally. To my taste this should for instance be mentioned (around line 235-240). Although clearly a possibility, I think other scenario's may apply as well here, so I suggest to change: 'is likely' (page 139) to: "could be". Also the claim that: "These results demonstrate that the PIF4-mediated thermo-responsive growth enhances plant survival of heat stresses in the natural environment" is too strong to my taste. I don't think these wetlab experiments are sufficient to claim relevance for the natural situation in the field and hence should be tuned down modestly.

The argument in the response letter that they did not include TOC1 and ELF4 before because "they regulate not only PIF4 but also other factors and thus their effects may be complex and not specific to PIF4 and thermo responses" is not limited to TOC1 and ELF4 only. Since PIF4 is also involved in many processes besides thermo-control of growth the same may apply to this factor. This could be briefly mentioned in the discussion.

Reviewer #2 (Remarks to the Author):

Although I do not find serious problems with the current version of the manuscript, some of the issues posed in my comments on the previous version are still present.

Now I understand what the authors wanted to conclude from the data presented in Fig. 6a, b, c. The only aim is to show that PIF4 increases seedling survival in response to heat stress. Although this is clearly supported by the data, I still find these observations disconnected from the rest for two reasons. First, as argued by the authors, this phenotype could be due to effects of PIF4 on plant morphology or on photosynthetic capacity. However, there are other explanations because plants acclimate to heat stress under warm temperatures via molecular processes that not necessarily involve changes in morphology or photosynthesis. Thus, we do not really know the mechanisms of the survival phenotype reported here. Second, the focus of the paper is on the control of thermomorphogenesis by the clock component TOC1 and these functional data do not specifically refer to the diurnal dependence of temperature cues.

As requested, the authors have added PIF4-only data to Fig. 1i. However, now it appears that PIF4 preferentially binds to promoters with multiple G-box motifs and that TOC1 can bind to promoters with only one G-box motif per 1 kb. It is not clear what this really tells us about the interaction between PIF4 and TOC1.

Figure format: There are still some figures that could be similar and have different format. For instance, Fig. 3 f and h should be in red for 29 C and blue for 20 C as Fig. 3g. No need to include two genes per box forcing the change of colour code.

Reviewer #3 (Remarks to the Author):

Overall, the manuscript is improved over the previous version. I accept the argument regarding membrane staining to show equal loading instead of blotting for a specific protein. The only remaining concern is the tenuous connection between the gating mechanism described here and thermotolerance. Figure 6 clearly shows PIF4 is important for a robust thermotolerance response and inhibition of its activity (either by mutation or TOC1-OX) leads to lower thermotolerance and enhanced PIF4 activity (PIF4-OX) promotes thermotolerance. These are novel and important observations. However, it remains unclear why gating of PIF4 activity is required to achieve strong thermotolerance. If the gating mechanism were adaptive for thermotolerance, then promotion of PIF4 activity at ZT0 would render WT more tolerant of heat shock during daytime than during nighttime, because nighttime is when the plant is unprepared for heat shock. WT in Figure 6A shows the opposite behavior. Instead, the more convincing argument is that external conditions dictate the effect of heat shock, where enhanced survival at ZT12 (in Figure 6a) is due to the absence of light immediately following the treatment, while the sensitivity at ZT0 arises from the morning light that follows treatment. In this case, clock-directed restriction of PIF4 activity to the morning is unnecessary. My recommendation is that Figure 6 remain in the manuscript, but the authors tone down the link between the gating mechanism and thermotolerance.

While improved, typos and grammatical errors remain throughout the text.

Reviewer #4 (Remarks to the Author):

This was a revised version of a manuscript previously submitted to Nature Communications. In this revised version the authors have addressed most of the concerns raised by the referees, with the exception of those regarding Figure 6 and its interpretation. This is unfortunate because these issues were raised independently by each of the referees.

The authors argue that the referees misunderstood the point that was made by this figure. Their revised argument is that Figure 6a shows that heat shock treatment is less damaging

to plants if followed by darkness, as plants treated at the beginning of the night survive better than plants treated in the morning. However these differences may equally be explained by circadian regulation. In order to test for this, plants should be heat-shock either in the morning or at dusk, and their survival should be compared following transfer to either light or darkness. If light enhances the effect of the heat shock, plants exposed to light should exhibit reduced survival, regardless of the time of the day when they were heat-shocked.

If the author's conclusion is correct and heat damage is enhanced by light, then it would make sense for plants to maximise their heat shock tolerance in the morning, but this is difficult to reconcile with the results in Figure 6a which show maximum survival following heat shock in the evening.

Figure 6b-c and supplementary Figure 9 show that PIF4 and TOC1 contribute to heat shock tolerance but it is unclear how this relates to the gating of temperature responses by the circadian clock.

While these results are interesting, more experiments would be required in order to clarify the link between heat shock tolerance and the circadian gating of high temperature responses. I agree with referee 2, that the data shown in figure 6 should be left out of the paper at this stage.

REVIEWERS' COMMENTS:

Reviewer #1 (Remarks to the Author):

The revised manuscript of Dr. Wang addresses most issues raised by the reviewers on a reasonable level by adding new data and amending the texts at several places.

The authors also significantly improved the texts by explaining the rationale behind testing heat stress tolerance in wild type and *pif4* mutant plants pre-subjected to control and warm temperatures (Fig 6). This part of the manuscript was confusing to all four reviewers and although improved, I suggest even more effort need to be taken to make crystal clear what justifies this twist in the flow of the paper. If well described it certainly has the potentially to enhance the quality paper, but if not it may appear an odd twist to those readers that are not entirely up to date with the state-of-the-art in the field of thermoresponsive growth.

I do agree with the notion of the authors in their response letter that a connection between high ambient pre-growth temperature was suggested before in literature, but never tested experimentally. To my taste this should for instance be mentioned (around line 235-240). Although clearly a possibility, I think other scenario's may apply as well here, so I suggest to change: 'is likely' (page 139) to: "could be". Also the claim that: "These results demonstrate that the PIF4-mediated thermo-responsive growth enhances plant survival of heat stresses in the natural environment" is too strong to my taste. I don't think these wetlab experiments are sufficient to claim relevance for the natural situation in the field and hence should be tuned down modestly.

The argument in the response letter that they did not include TOC1 and ELF4 before because "they regulate not only PIF4 but also other factors and thus their effects may be complex and not specific to PIF4 and thermo responses" is not limited to TOC1 and ELF4 only. Since PIF4 is also involved in many processes besides thermo-control of growth the same may apply to this factor. This could be briefly mentioned in the discussion.

Response:

We have removed the confusing Fig 6a, and moved Supplementary Fig 9 to Fig 6c. Following the suggestions of this reviewer, we have further revised the paragraph describing Fig 6, to clarify that the adaptive benefit of PIF4-mediated thermomorphogenesis has been speculated, but never tested. We hope the new paragraph now makes it clear that the experiments were designed to confirm that pre-exposure to warm temperature increases heat tolerance, and such adaptation is mediated by PIF4 and negatively regulated by TOC1 (heat tolerance data of *toc1;prr5* and *TOC1-OX* is in Fig 6c). These results confirm the functions of PIF4 and TOC1 in regulating thermo-adaptation and survival of heat stress, which occurs around noon in nature. Such functions are consistent with the maximum temperature sensitivity of PIF4 in the morning, provided by the circadian rhythm of TOC1. We have added thermo-tolerance data of *TOC1-OX* and *toc1;prr5*, which is consistent with the PIF4 data. These results together provide strong support for the model that the PIF4 activation by day-time warm temperature, allowed by the trough level of TOC1, enhances thermomorphogenesis and plant survival of heat stresses which normally occur around noon.

Reviewer #2 (Remarks to the Author):

Although I do not find serious problems with the current version of the manuscript, some of the issues posed in my comments on the previous version are still present.

Now I understand what the authors wanted to conclude from the data presented in Fig. 6a, b, c. The only aim is to show that PIF4 increases seedling survival in response to heat stress. Although this is clearly supported by the data, I still find these observations disconnected from the rest for two reasons. First, as argued by the authors, this phenotype could be due to effects of PIF4 on plant morphology or on photosynthetic capacity. However, there are other explanations because plants acclimate to heat stress under warm temperatures via molecular processes that not necessarily involve changes in morphology or photosynthesis. Thus, we do not really know the mechanisms of the survival phenotype reported here. Second, the focus of the paper is on the control of thermomorphogenesis by the clock component TOC1 and these functional data do not specifically refer to the diurnal dependence of temperature cues.

Response: First, our discussion of the two possible contributors to the phenotypes is based on previous reports, which we cited. We do not discuss other possibilities that have no supporting evidence, because adaptation should involve complex changes at biochemical, physiological and morphological levels. However, our results are consistent with PIF4's effect on plant morphology and photosynthetic capacity being at least part of the mechanisms. Second, genetic evidence for the functions of PIF4 and TOC1 in regulating heat adaptation is very important in this study, because our model predicts maximum thermoresponsiveness around morning-noon before the hottest time of day. Such timing is consistent with adaptation to heat stress, in contrast to an evening/night-specific thermos-response, which may have other physiological functions unrelated to the heat stress imposed by the hottest temperature of the day.

As requested, the authors have added PIF4-only data to Fig. 1i. However, now it appears that PIF4 preferentially binds to promoters with multiple G-box motifs and that TOC1 can bind to promoters with only one G-box motif per 1 kb. It is not clear what this really tells us about the interaction between PIF4 and TOC1.

Response: Our label in Fig 1i was misleading. We have changed the y-axis label to "Frequency of motif per 1 kb". The frequency of G-box motif indicates relative enrichment of the motif in the binding sequences identified in ChIP-seq, not absolute number of the motif per promoter. The data shows that TOC1 tends to bind to PIF4-TOC1 shared targets through G-Box but binds to non-PIF4 targets independent of G-box, which is consistent with TOC1 being recruited to the shared promoters by PIF4, which binds to G-box.

Figure format: There are still some figures that could be similar and have different format. For instance, Fig. 3 f and h should be in red for 29 C and blue for 20 C as Fig. 3g. No need to include two genes per box forcing the change of colour code.

Response: We have changed Fig. 3f and 3h format to make the graph color codes consistent in all the figures, i.e. red for 29°C and blue for 20°C. Specifically, in Fig. 3f, we have removed graphs for *IAA19* since the same results have been shown in supplemental fig. 3; in Fig. 3h, we have put *YUC 8* and *IAA20* gene names on x-axis label, so we have kept two genes in one chart with the same color codes as the other figures.

Reviewer #3 (Remarks to the Author):

Overall, the manuscript is improved over the previous version. I accept the argument regarding membrane staining to show equal loading instead of blotting for a specific protein. The only remaining concern is the tenuous connection between the gating mechanism described here and thermotolerance. Figure 6 clearly shows PIF4 is important for a robust thermotolerance response and inhibition of its activity (either by mutation or TOC1-OX) leads to lower thermotolerance and enhanced PIF4 activity (PIF4-OX) promotes thermotolerance. These are novel and important observations. However, it remains unclear why gating of PIF4 activity is required to achieve strong thermotolerance. If the gating mechanism were adaptive for thermotolerance, then promotion of PIF4 activity at ZT0 would render WT more tolerant of heat shock during daytime than during nighttime, because nighttime is when the plant is unprepared for heat shock. WT in Figure 6A shows the opposite behavior. Instead, the more convincing argument is that external conditions dictate the effect of heat shock, where enhanced survival at ZT12 (in Figure 6a) is due to the absence of light immediately following the treatment, while the sensitivity at ZT0 arises from the morning light that follows treatment. In this case, clock-directed restriction of PIF4 activity to the morning is unnecessary. My recommendation is that Figure 6 remain in the manuscript, but the authors tone down the link between the gating mechanism and thermotolerance.

Response: We have removed Fig 6a, which confused every reviewer. But to clarify again, if PIF4 activity is restricted to the night instead of morning, plant would grow a morphology based on the night temperature, and such morphology may not be fit for the warmest temperature which occurs around noon. The benefit of PIF4 is predicted to be due to architectural changes not immediate responses. We have revised the description and discussion of Fig 6 according to comments of all reviewers. Overall, the link between gating mechanism and thermotolerance has been toned down (e.g. we use “suggest” instead of “demonstrate” in the last sentence).

While improved, typos and grammatical errors remain throughout the text.

Response: We have improved the language and revised the manuscript according to the journal policies and format requirements.

Reviewer #4 (Remarks to the Author):

This was a revised version of a manuscript previously submitted to Nature Communications. In this revised version the authors have addressed most of the concerns raised by the referees, with the exception of those regarding Figure 6 and its interpretation. This is unfortunate because these issues were raised independently by each of the referees.

The authors argue that the referees misunderstood the point that was made by this figure. Their revised argument is that Figure 6a shows that heat shock treatment is less damaging to plants if followed by darkness, as plants treated at the beginning of the night survive better than plants treated in the morning. However these differences may equally be explained by circadian regulation. In order to test for this, plants should be heat-shock either in the morning or at dusk, and their survival should be compared following transfer to either light or darkness. If light enhances the effect of the heat shock, plants exposed to light should exhibit reduced survival, regardless of the time of the day when they were heat-shocked.

If the author's conclusion is correct and heat damage is enhanced by light, then it would make sense for plants to maximise their heat shock tolerance in the morning, but this is difficult to reconcile with the results in Figure 6a which show maximum survival following heat shock in the evening.

Response: Sorry for the confusion. Our interpretation of the results is that maximum survival is observed following the evening heat shock because heat shock followed by darkness is not damaging due to requirement of light for heat damage, not because plants are better prepared for heat at evening. Plants maximize their heat shock tolerance through PIF4-mediated adaptation response, but stress with maximum adaptation (morning) is still worse than no stress (evening). Further, there was no adaptation treatment in this experiment, and therefore the experiment only compares damaging effects of heat followed by light vs dark. The point is that plants only need to deal with heat stress during the day, which is obvious for many other reasons. Therefore, we have removed Fig 6a.

Figure 6b-c and supplementary Figure 9 show that PIF4 and TOC1 contribute to heat shock tolerance but it is unclear how this relates to the gating of temperature responses by the circadian clock.

While these results are interesting, more experiments would be required in order to clarify the link between heat shock tolerance and the circadian gating of high temperature responses. I agree with referee 2, that the data shown in figure 6 should be left out of the paper at this stage.

Response: We have removed Figure 6a, as most reviewers were confused about the difference between survival due to lack of stress and presence of adaptation. However, the data in Figure 6b and 6c (and supplementary Figure 9) provides important genetic

evidence for the function of PIF4 and suppression of TOC1 level in thermotolerance. As Reviewer 3 pointed out, “these are novel and important observations”. Such functions in thermotolerance are consistent with the circadian timing of PIF4 activation (morning to noon) relative to the external timing of heat stress (noon to early afternoon) in nature. In other words, if PIF4’s function is to mediate a response to warm evening temperature, such as the flowering regulation by night-time temperature (Kinmonth-Schultz et al., 2016), it may not be related to heat stress tolerance. Therefore, we decide to move previous Supplementary Figure 9 showing thermotolerance of *TOC1-OX* and *toc1;prr5* to Figure 6c. We also have changed the text accordingly to clarify the aim of Fig 6 in the last paragraphs of Results and Discussion.